# Corticothalamic neurons in motor cortex have a permissive role in motor execution

Lina Marcela Carmona[1], Anders Nelson[2], Lin T. Tun[1], An Kim[1], Rani Shiao[3], Michael D. Kissner[4], Vilas Menon [5] & Rui M. Costa [1,6] ✉

The primary motor cortex (M1) is a central hub for motor learning and execution. M1 is composed of heterogeneous cell types with varying relationships to movement. Here, we tagged active neurons at different stages of motor task performance in mice and characterized cell type composition. We identified corticothalamic neurons (M1$^{CT}$) as consistently enriched with training progression. Using two-photon calcium imaging, we found that M1$^{CT}$ activity is largely suppressed during movement, and this negative correlation augments with training. Increasing M1$^{CT}$ activity through closed-loop optogenetic manipulations during forelimb movement significantly hinders execution, an effect that became stronger with training. Similar manipulations, however, had little effect on locomotion. In contrast, M1 corticospinal neurons positively correlate with movement, with an increase during training. We uncovered that M1$^{CT}$ neurons suppress corticospinal activity via feedforward inhibition, also scaling with training. These results identify a permissive role of corticothalamic neurons in movement execution through disinhibition of corticospinal neurons.

The ability to learn and execute new movements is critical for survival. Precise execution of movements is often acquired via repetition and refinement, a process that depends on activity coordinated throughout the brain and spinal cord[1,2]. Within these circuits, the primary motor cortex (M1) projects directly to the spinal cord, but also to other motor areas in the basal ganglia, thalamus and brainstem, affording it unique control over motor action[3,4]. M1 is engaged during motor learning as well as during the execution of voluntary movements in rodents, non-human primates, and humans[5–12]. Changes in the formation, turnover, and spatial distribution of spines[13–17], the restructuring of dendritic arbors[18–21], as well as shifts in the patterns of electrical activity in M1 all affirm an active and critical role of M1 in learning and performance of various motor tasks[11,15,22–27].

However, M1 cells are transcriptionally and functionally heterogeneous[28–30], and different cell types exhibit varying relationships to movement during learning and/or execution. Furthermore, many of these populations are intricately connected, generating local networks that could contribute to the overall activity and function of M1. Some recent studies have tried to isolate the role of different M1 cell types in movement or learning by using known cell type markers or projection patterns[22,23,31]. We reasoned that a more unbiased and comprehensive screen of which cell types are engaged during movement, at different stages of training and proficiency, could complement the candidate approach taken to date. We designed an experimental approach leveraging a calcium-dependent photoswitchable indicator in combination with single-cell RNA sequencing to determine which M1 cell types are differentially enriched at different stages of motor training and proficiency. This approach identified several neuron types within M1 with differential activity early versus late in training and led us to uncover that a particular cell type—the M1 corticothalamic neurons—have an unexpected but critical role in movement.

[1]Department of Neuroscience, Zuckerman Mind Brain Behavior Institute, Columbia University, New York, NY, USA. [2]Center for Neural Science, New York University, New York, NY, USA. [3]Laboratory of Molecular Genetics, The Rockefeller University, New York, NY, USA. [4]Columbia Stem Cell Initiative, Columbia University Irving Medical Center, New York, NY, USA. [5]Center for Translational & Computational Neuroimmunology, Department of Neurology, Columbia University Irving Medical Center, New York, NY, USA. [6]Allen Institute, Seattle, WA, USA. ✉e-mail: rui.costa@alleninstitute.org

## Results

### M1<sup>CT</sup> neurons are enriched at late stages of training

We utilized an M1-dependent head-fixed motor task[32] to screen for cell type engagement at different stages of training. The task consists of forelimb-driven pulls of a small rotary wheel wherein mice had to cross an infrared (IR) beam at the apex of the wheel and make contact with the rungs to commence a trial (Fig. 1A). Mice were rewarded when pulls executed within 200 ms of trial start reached a velocity threshold.

Within two weeks, mice showed an increase in the number of successful trials, as demonstrated by significantly increased percent success over time (p = 0.0104, one-way ANOVA, time) (Fig. 1B), as well as increased velocity of trials, as indicated by an increase in the maximum velocity reached during all trials in each training session (p = 0.0182, one-way ANOVA, time) (Fig. 1C, D). For subsequent experiments, we chose day 4 and day 12 as our early and late training time points, respectively (p = 0.0336, day 4 v. day 12, one-way ANOVA).

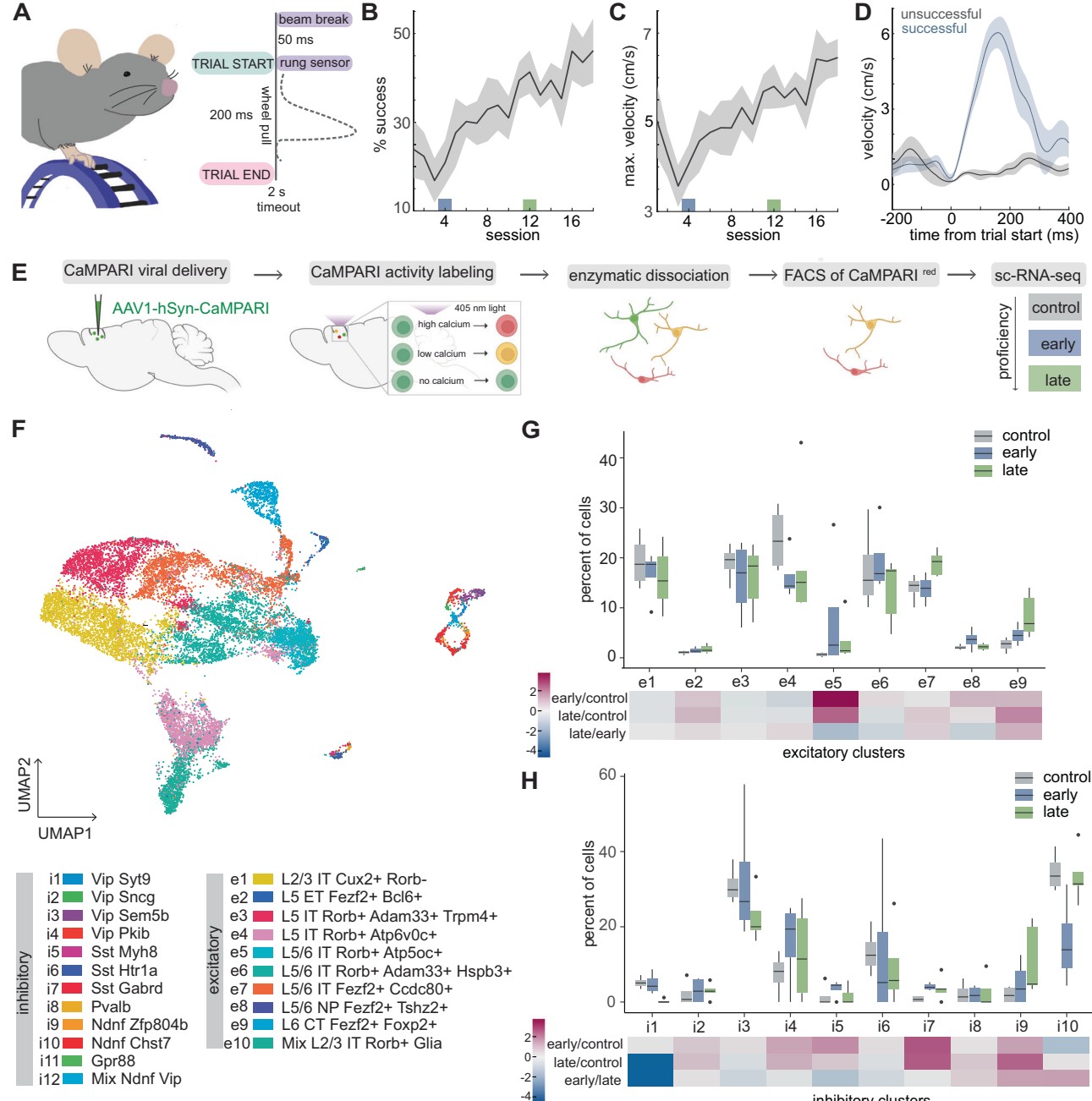

**Fig. 1 | M1 active cell type enrichment during training of a motor task.**
**A** Schematic of wheel pulling task, left. Schematic of trial structure, right. Percent success (**B**) and maximum velocity (cm/s) of all trials (**C**) during 18 consecutive training sessions; N = 6 animals. Mean, solid gray line; shaded area, SEM; Early and late training are denoted by blue and green boxes, respectively; p = 0.0104 (**B**) and p = 0.0182 (**C**), one way ANOVA, time; p = 0.0336 (**B**) and p = 0.0569 (**C**) early v. late training, one-way ANOVA multiple comparisons; see Supplementary Table 3 for all statistics. **D** Velocity (cm/s) of successful (blue) and unsuccessful trials (gray) on day 4 (early training);

mean, solid line; shaded area, SEM. **E** Outline of labeling, sorting, and sc-RNAseq strategy; created in BioRender. Carmona (2025) https://BioRender.com/t64l613. **F** Aggregate UMAP of all sc-RNA-seq runs. Percent of excitatory (**G**) or inhibitory (**H**) neurons present from each clusters at each timepoint, top; Gray, control; Blue, early training; Green, late training; outliers, black dots. Fold change between the early and control, late and control, and early and late, bottom; N = 4, control; N = 4, early; N = 5, late. Each of these replicates consists of neurons pooled from 2-4 animals; see Supplementary Table 2 for statistics of all comparisons.

We next optimized a system to label the neurons that were active at different stages of execution of this task. To this end, we utilized CaMPARI, a photoswitchable indicator that fluoresces green at baseline but converts to red upon coincident exposure to calcium and 405 nm light[33] (Fig. 1E). Although there are numerous systems now available to label active neurons[34–37], CaMPARI was particularly well suited for our approach given its high fidelity reporting of activity relative to Immediate Early Gene (IEG) systems, and its short window of labeling (on the order of minutes) compared to hours for other systems. We performed stereotactic injections of an adeno-associated virus (AAV) expressing CaMPARI under the neuron-specific human synapsin promoter into the caudal forelimb region[38,39] of M1. We compared cell type engagement at different training stages: 1) early stages of training (day 4), 2) late stages of training (day 12), and 3) no training, in which the wheel was covered but the mouse was otherwise exposed to the same environment (collected on the equivalent of day 1 of a training timeline). At the respective time point, light was delivered through a cranial window in pulses of 5 s to photoswitch CaMPARI and label the active neurons as mice performed or were exposed to the task (Fig. 1S). Labeling epochs were random and irrespective of specific behavior during the training session. M1 was then rapidly dissociated and neurons with photoswitched red CaMPARI fluorescence were enriched using fluorescence activated cell sorting (FACS). This ensured that only active neurons were subsequently processed using single-cell RNA sequencing. Each timepoint (control, early, late) was repeated a minimum of four times to account for behavioral variability, with each run consisting of neurons from 2-4 animals. We achieved a median unique molecular identifier (UMI) and gene count of 7971.5 and 3503, respectively, over all runs (Supplementary Table 1).

Unbiased clustering of our aggregate data yielded the expected cell type subdivisions based on cortical layer marker genes. Mapping to a high resolution M1 cell type atlas further demonstrated the cell type heterogeneity present in our samples, with concordance among our clusters and published annotations[28,29] (Figs. 1F and S2A, B). Three clusters, i11, i12, and e10, contained mixed populations and were excluded from further analysis. Comparison across each timepoint did not reveal a binary presence or absence of particular cell types (Fig. S2C). Rather, enrichment analysis demonstrated a diversity of cell type engagement patterns across timepoints (Fig. 1G, H). Because we did not capture as many inhibitory neurons across timepoints, we conducted our enrichment analysis independently for excitatory and inhibitory neurons. To control for labeling dependency on light penetration and differential baseline activity of neuronal types, we used internal comparisons by calculating enrichment relative to the control condition that did not receive any training. Overall, compared to the control condition that did not receive any training, the most enrichment occurred across clusters in late condition with three subtypes enriched in excitatory neurons (e2, e5, e7, and e9) and three subtypes in inhibitory neurons (i2, i7, and i9). In comparison, only one inhibitory (i7) subtype was enriched in the early condition relative to the control condition. When comparing early and late training stages, we observed enrichment of three subtypes, one excitatory (e6) and one inhibitory (i10), at late stages of training, and enrichment of one inhibitory subtype (i1) at early stage (significance determined using ANCOM-BC p value <0.05; Supplementary Table 2). One subtype of particular interest with significant enrichment at the late training timepoint relative to the control (p = 0.00394) because of its lack of characterization during motor learning and execution was the cluster of corticothalamic neurons (M1$^{CT}$). This population is marked by FoxP2 and Fezf2 expression (excitatory cluster 9, e9) and is experimentally accessible with the FoxP2-cre mouse line. Anatomical mapping of the projections of FoxP2 + M1 neurons confirmed broad thalamic targeting of this population[4,40–42], and also demonstrated topographical organization in M1 dictated by thalamic projection pattern (Fig. S3). Given this enrichment at late stages of learning, as well as the known role of

thalamus in motor learning and execution[43–45], we further investigated the role of these neurons during motor execution.

## M1$^{CT}$ neurons decrease activity during movement

The contribution of the corticothalamic M1 output pathway to motor execution is poorly understood compared to the thalamocortical pathway[46,47] as well as other output populations in M1[23,27,31]. Thus, we characterized the activity of M1$^{CT}$ neurons at different stages of training in our wheel pulling task. We performed stereotactic injections to deliver an AAV expressing Cre-dependent GCaMP7f to the caudal forelimb area in M1 of FoxP2-cre mice (Fig. S4A, B). To image activity of these deep cortical neurons without incurring significant damage to M1, we followed an approach previously implemented to image corticospinal neurons in M1[23,27]. We used two-photon (2p) imaging to record calcium dynamics in the dendritic trunks of FoxP2 + M1$^{CT}$ neurons through a cranial window at least 300 μm below pia. An additional retrograde AAV expressing a Cre dependent red fluorescent protein was injected into the cervical segments of the spinal cord to label the small population of Foxp2 expressing corticospinal neurons and exclude them from our imaging fields (Fig. S4A–C). To extract the calcium signal, we used constrained nonnegative matrix factorization (CNMF)[48], and we excluded any highly correlated units (ρ > 0.8) to avoid overrepresentation from branches of the same neurons[23,49].

We analyzed M1$^{CT}$ neuronal activity during movement by aligning to trial start during successful or unsuccessful trials across all training sessions. Surprisingly, we observed that activity of M1$^{CT}$ decreased during wheel movement (Figs. 2A, B and S4E). To examine whether this was a general feature of these neurons, we calculated the cross-correlation of the activity (ΔF/F) to the wheel velocity as measured by the rotary encoder. This further demonstrated a negative relationship between activity and wheel movement throughout the session, with the suppression in neuronal activity slightly preceding peaks in wheel velocity (Fig. 2C). No correlation was observed when the neuronal activity shuffled (Fig. S4D). Given this striking signature in M1 where other populations exhibit positive correlations with movement[23,27,31], we examined whether the suppression in M1$^{CT}$ neurons was observed for the majority of M1$^{CT}$ neurons. For this, we examined activity during all wheel pulls of all sessions as we found a similar decrease in activity during movement as in trials (Fig. S4F). Classification revealed that most neurons show decreased activity during wheel pulls (69%, M1$^{CT-down}$) relative to a window of low movement prior to the pull. A smaller fraction displayed no modulation (15%, M1$^{CT-no mod.}$) or increased activity (16%, M1$^{CT-up}$) during wheel movement (Figs. 2D–G and S4G). To further validate this finding, we performed similar imaging using another well-established marker line for labeling layer VI corticothalamic neurons, Ntsr1-cre (Fig. S4H, I). Classification of modulation corroborated that the majority of M1$^{CT}$ neurons decreased in activity during wheel pulls (Fig. S4I).

Furthermore, for the neurons that decreased in activity, the magnitude of the decrease changed depending on the velocity of the movement. When we grouped pulls of varying velocity into bins (2–6 cm/s, 6–10 cm/s, 10–14 cm/s, >14 cm/s), we observed larger decreases in activity for higher velocity wheel movements (Fig. 2H–K; see Supplementary Table 3 for summary of statistics; n = 740 units, N = 4 mice; Fig. S5B, C), but no change in the timing of maximal suppression (Fig. S5A).

A decrease in activity was an unexpected finding, as we had observed increases in the enrichment of this population over training in our activity-based sequencing screen (Fig. 1G). However, during our initial screen, the labeling with CAMPARI was performed at random times in the session. We therefore hypothesized that the activity of M1$^{CT}$ neurons late in training was higher during non-movement periods and lower in movement periods. We tested this idea by performing closed-loop labeling of CaMPARI in M1$^{CT}$ neurons to enrich for periods of movement or no movement. We performed stereotactic injections

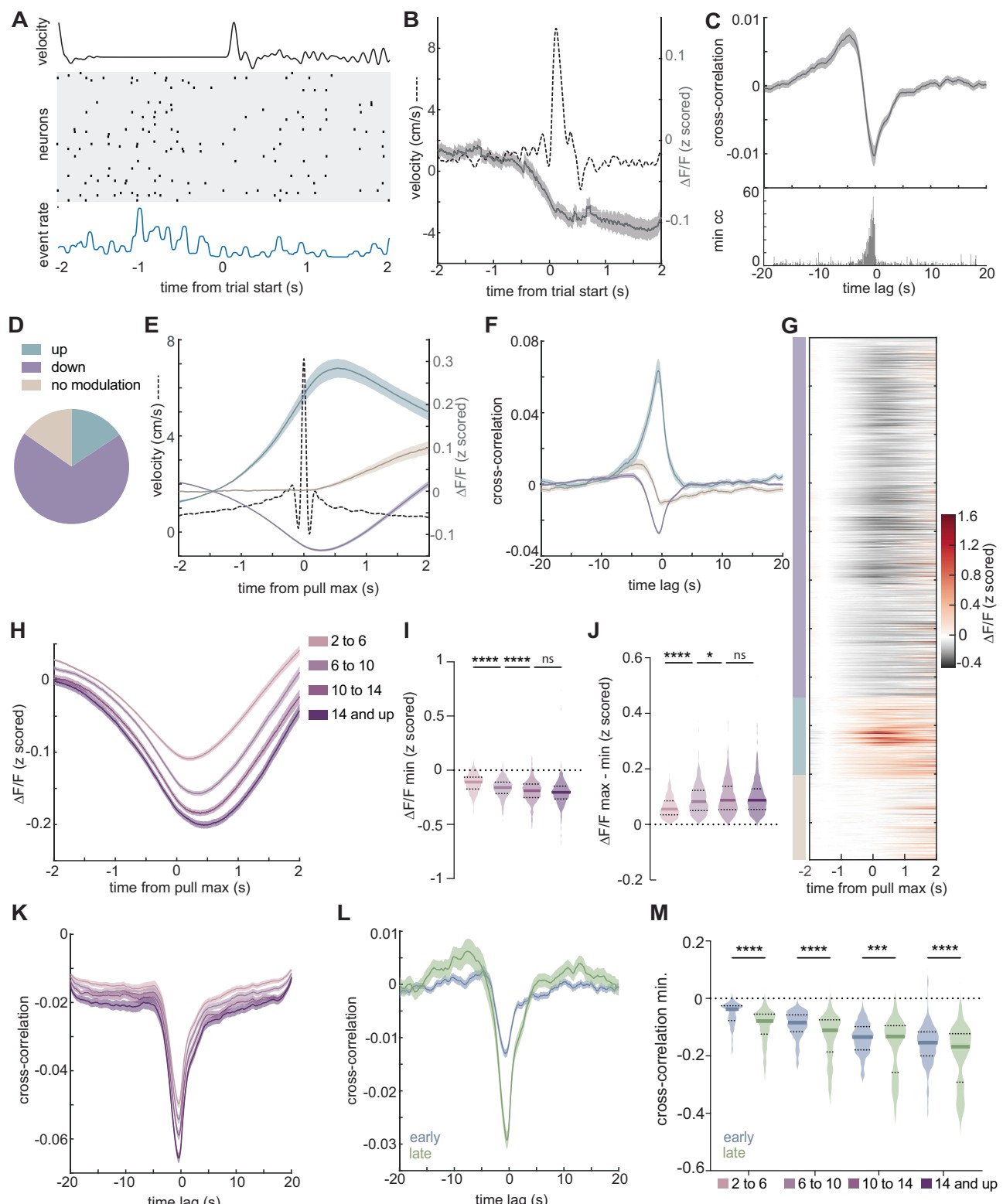

of an adeno-associated virus (AAV) expressing cre dependent CaM-PARI under the neuron-specific human synapsin promoter into the caudal forelimb region[38,39] of M1 in Ntsr1-cre mice. During training, we delivered 405 nm photoconversion light for 500 ms either during movement (trials ON) or during periods excluding trials (trials OFF) (Fig. S6A, B). Confirming our hypothesis, we observed that significantly more neurons were labeled outside of trial periods than during trials/movement (Fig. S6C; p = 0.0068, two-tailed t-test). This suggests that during training M1^CT neurons are more active in periods of no

movement. Furthermore, we observed an increase in the modulation range (ΔF/F max – min) of neurons around movement from early to late training (Fig. S6D). We therefore examined if the correlation also changed over the span of training sessions and observed a significantly larger anti-correlation during late training sessions compared to early (Fig. 2L, M) in M1^CT-down neurons. No difference was observed in the relative proportions of the population (Fig. S7A). Given that as mice progressed through the task, they tended to perform higher velocity pulls (Figs. 1C and S7B), we examined the anti-correlation in early and

**Fig. 2 | M1$^{CT}$ neurons are suppressed during movement. A** Single trial example of M1$^{CT}$ activity; top, wheel velocity; middle; raster plot (N = 53 units); bottom, aggregate event rate; **B** Z-scored ΔF/F during successful trials, all sessions; mean, solid line; shaded area, SEM; Wheel velocity, dotted line; **C** Top, cross-correlation of Z-scored ΔF/F by unit to wheel velocity, whole session; mean, solid line; shaded area, SEM; bottom, binned cross-correlation minimum; 50 ms bins. **D** Unit classification on activity during wheel pulls. Z-scored ΔF/F, wheel pulls (**E**) and cross-correlation to wheel velocity, whole session (**F**) by group; solid lines, mean; shaded areas, SEM; **G** Z-scored ΔF/F of units over wheel pulls; groups denoted on left. **H** Z-scored ΔF/F, wheel pulls of increasing velocities (cm/s), down modulated neurons; solid line, mean; shaded area, SEM. Distribution of z-scored ΔF/F minimum, wheel pulls (**I**) or difference in z-scored ΔF/F maximum and minimum before and during wheel pulls (**J**), down modulated neurons; colors as in (**H**). Thick line, mean; thin lines, quartiles. One-way ANOVA, multiple comparisons; $p < 1 \times 10^{-15}$, 2 to 6 v. 6 to 10 and 6 to 10 v. 10 to 14; p = 0.0553, 10 to 14 v. 14 and up (**I**), $p < 1 \times 10^{-15}$, 2 to 6 v 6 to 10; p = 0.230, 6 to 10 v 10 to 14; p = 0.994, 10 to 14 v 14 and up (**J**); see Supplementary Table 3 for summary of statistics. Cross-correlation to wheel velocity, trials (**K**) and whole session (**L**) for wheel pulls by velocity group (**K**) or early (blue) and late (green) sessions (**L**), down modulated neurons; solid line, mean; shaded area, SEM. **M** Distribution of minimum cross-correlation to wheel velocity, pulls of increasing velocity in early (blue) or late (green) sessions, down modulated neurons; thick, line, mean; thin lines, quartiles. Early v. late comparisons, one-way ANOVA, multiple comparisons, $p = 4.14 \times 10^{-8}$, 2 to 6, $p = 3.89 \times 10^{-9}$ 6 to 10, p = 0.005, 10 to 14, and $p = 2.71 \times 10^{-7}$ 14 and up; see Supplementary Table 3 for summary of statistics. For **B**, **C**, n = 1078 units from N = 4 mice. For **D**–**G**, n = 740 downmodulated units, n = 168 upmodulated units, n = 168 non-modulated units, all from N = 4 mice. For **L**, **M**, early: n = 231 units, late: n = 228 units from N = 4 mice.

late session neurons for pulls of matched velocities. Even in subsets of pulls with a similar distribution of peak velocities (Fig. S7B), we found a consistent increase in the magnitude of the anti-correlation to wheel movement at late training (Figs. 2M and S7C; see Supplementary Table 3 for summary of statistics; n = 231 units, early, n = 228 units, late, N = 4 mice). These results indicate that M1$^{CT}$ neurons are predominantly suppressed during movement and show a negative correlation with movement that is modulated by speed and increases with training. Given their increased representation in the active population over training, we sought to understand whether this decreased activity was necessary for movement execution.

## M1$^{CT}$ suppression is permissive for execution of learned movement

We hypothesized that the decrease in M1$^{CT}$ neuronal activity was critical for movement execution during training, and that disrupting this decrease in activity would affect movement. To test this, we performed a closed-loop optogenetic manipulation on a subset of trials at the late training stage where we identified these neurons as most enriched. Given that M1$^{CT}$ neurons decrease in activity during wheel pulls, we chose to activate them during this time window to disrupt this expected decrease. To achieve this, we performed stereotactic injections of an AAV into the caudal forelimb area of M1 to express Cre-dependent channelrhodopsin (hChR2) or a fluorescent protein in Foxp2-cre animals, generating our opsin and control groups respectively (Fig. S8A). We then trained both groups of animals to the late training stage and delivered 400 ms of pulsed light (20 hz; 10 ms pulse width) at the onset of one third of trials (Fig. 3A). Overall, this manipulation decreased the number of successful trials performed with the light on in opsin expressing animals but not in control animals (Fig. 3B, C; for percent of successful trials with light on: # successful trials during light on / # of all successful trials, p = 0.0119, two-tailed unpaired t-test; for percent success: # successful trials during light on (OR off) / # of all trials with light on (OR off), p = 0.7031 and p = 0.0014 for control and hChR2, respectively, see Supplementary Table 3 for summary of all other statistics; N = 7 mice, control, N = 6 mice, hChR2). Importantly, when we compared the way in which trials were executed with or without light, we found a drastic decrease in wheel velocity (p = 0.871 and p = 0.0003 for control and hChR2, respectively, two-way repeated measures ANOVA; N = 7 mice, control, N = 6 mice, hChR2; see Supplementary Table 3 for summary of all statistics) as well as pull distance (p = 0.951 and p = 0.0027 for control and hChR2, respectively, two-way repeated measures ANOVA; N = 7 mice, control, N = 6 mice, hChR2; see Supplementary Table 3 for summary of all statistics) in light-on trials of opsin animals (Figs. 3D–F and S8B–G). This deficit was also reflected in a decrease of the maximum distance traveled by the right forelimb during trials of opsin animals relative to control animals (Fig. S8H, p = 0.195 and p = 0.0088 for control and hChR2 respectively, two-way repeated measures ANOVA; N = 7, control group; N = 6, hChR2 group; see Supplementary Table 3 for summary of all statistics;

Supplementary Movies 1–4). Furthermore, this decrease in velocity was rapid, and observed in less than 200 ms (Fig. 3D; p < 0.0001 and p > 0.999 for opsin and control, respectively; two-way ANOVA for 400 ms after light on, time x condition). Again, we validated this finding by repeating the same manipulation in the Ntsr1-cre mouse line. We observed a similar deficit although it was less pronounced by certain metrics (Fig. S9).

Our imaging experiments revealed that the activity of M1$^{CT}$ neurons had lower magnitude anti-correlation to movement early in training relative to late training (Fig. 2L, M). Consistently, our sc-RNAseq screen detected a gradual enrichment of this cell type with more enrichment in late training than in early training (Fig. 1G). Therefore, we tested whether this activity signature was critical for forelimb movement early in training. As before, we set up a cohort of opsin and control FoxP2-cre animals but trained them to the early learning stage and delivered light concurrent with trial start, (20 hz; 10 ms pulse width; Fig. 3G). Overall, the effect of this manipulation on movement execution was weaker with a non-significant decrease in the number of successful trials performed with light on in opsin expressing animals relative to control animals compared to the late training manipulations (Fig. 3H, I; for percent of successful trials with light on: # successful trials during light on / # of all successful trials, p = 0.0933, two-tailed unpaired t-test; for percent success: # successful trials during light on (OR off) / # of all trials with light on (OR off), p = 0.977 and p = 0.102 for control and hChR2, respectively, see Supplementary Table 3 for summary of all other statistics; N = 5 mice, control, N = 6 mice, hChR2). The most apparent deficit was observed when comparing how trials were executed (Figs. 3J, K and S10A, B, D, E) where we noted a decrease in wheel velocity in light trials only in hChR2 animals (for velocity trace: p < 0.0001 and p > 0.9999 for opsin and control, respectively; two-way ANOVA for 400 ms after light on, time x condition; for mean max velocity: p = 0.981 and p = 0.0331 for control and hChR2, respectively, see Supplementary Table 3 for summary of all other statistics; N = 5 mice, control, N = 6 mice, hChR2) although this did not extend to total pull distance (Figs. 3L and S10C–F; p = 0.592 and p = 0.191 for control and hChR2, respectively, see Supplementary Table 3 for summary of all other statistics). Together, these data reveal that activating M1$^{CT}$ neurons at precise movement times when their activity normally decreases rapidly impairs movement execution, and strongly suggest that the decrease in activity of M1$^{CT}$ neurons is permissive for this learned movement, especially late in learning.

## Significance of M1$^{CT}$ suppression is dependent on type of movement

Whereas these results showed that disruption of the activity of these neurons impaired proficient execution of task-related movements, they did not indicate whether modulation of M1$^{CT}$ neuronal activity is important for ongoing movement after peak velocity has been achieved. We, therefore, performed a closed-loop manipulation on a new cohort of animals, delivering light around the time of maximum

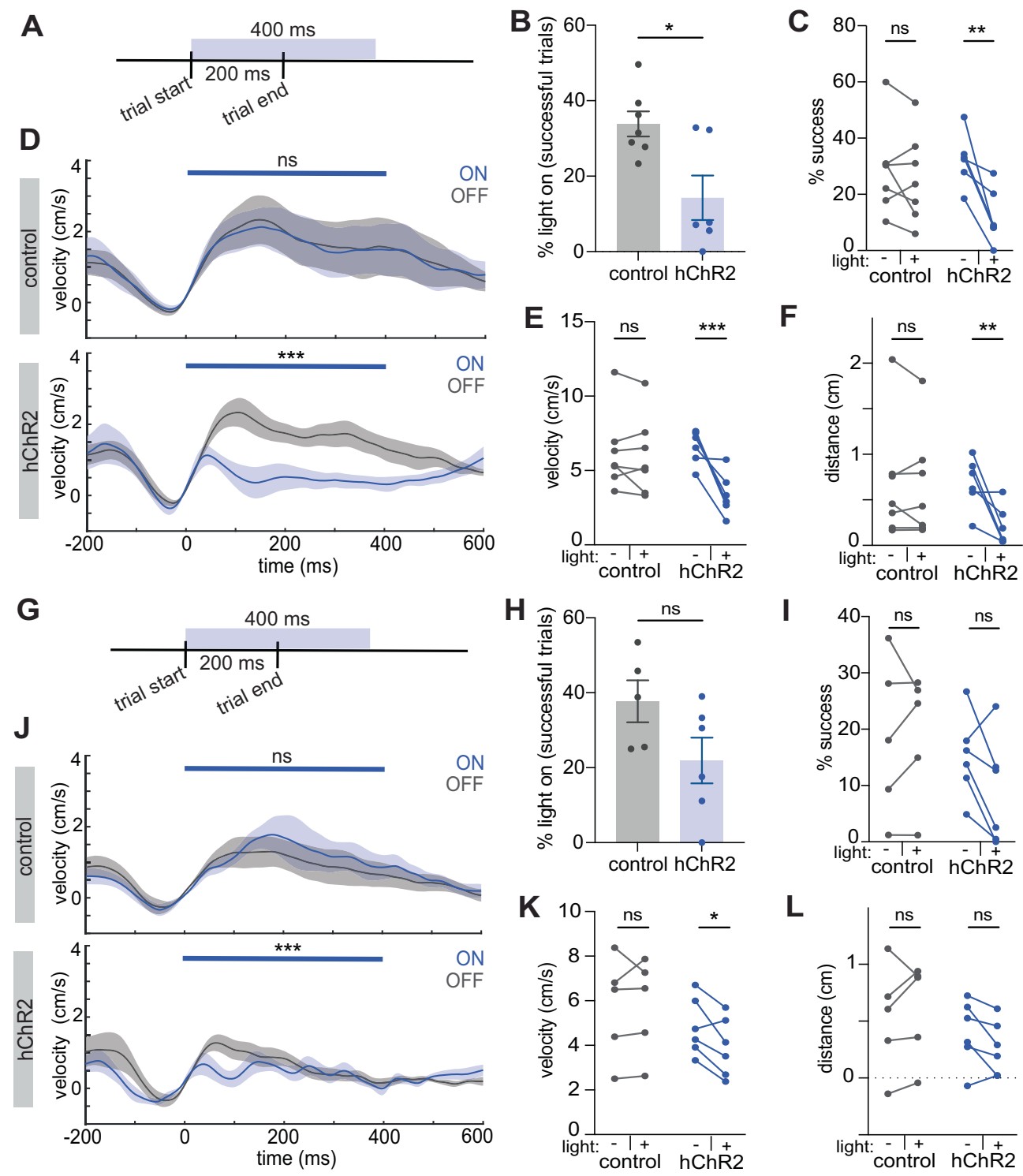

trial velocity, rather than trial initiation at late training (Fig. 4A, light delivered 110 ms after trial start). As expected, there was no change in the percent of successful trials (p = 0.164, two-tailed unpaired t-test; N = 6 mice, control, N = 6 mice, hChR2) or maximum velocity of trials (see Supplementary Table 3 for summary of all statistics, N = 6 mice, control, N = 6 mice, hChR2) as mice had already reached peak velocity before the light was turned on (Figs. 4C and S11B, C). However, we observed that even after movement had started, there was a rapid and drastic decrease in movement speed following stimulation (Figs. 4B, D and S11A, D–I). Finally, we tested whether stimulating these neurons before movement initiation (i.e. to prevent the decrease in M1$^{CT}$ neural

activity before movement start) would affect task execution. We performed an open loop optogenetic manipulation, randomly delivering pulsed light (20 hz; 10 ms pulse width) for 1200 ms over 1/10$^{th}$ of the behavior session (Fig. S12A). When we examined wheel pulls similar to those executed during trials but where light was delivered before trial initiation (50–75 ms before the peak of the pull), we observed a decrease in both velocity (p = 0.156 and p = 0.0016 for control and hChR2, respectively, two-way repeated measures ANOVA, see Supplementary Table 3 for summary of all statistics; N = 11 mice, control, N = 11 mice, hChR2) and wheel pull distance (Fig. S12B–D; p = 0.0708 and p = 0.0011 for control and hChR2, respectively, two-way repeated

**Fig. 3 | M1^CT activity disruption perturbs movement execution.** Schematic of optogenetic light delivery for closed loop at trial start at late training (**A**) and early training (**G**) for wheel turning task. Percent of successful trials with light on (# successful trials during light on / # of all successful trials) at late training (**B**) and early training (**H**); control, gray; hChR2, blue; error bars, SEM; two-tailed unpaired t-test, p = 0.0119 for (**B**) and p = 0.0933 for (**H**). For **B**–**F**: N = 7 animals, control group; N = 6 animals, hChR2 group. For **H**–**L**: N = 5 animals, control group; N = 6 animals, hChR2 group. Each point is average of all light delivery sessions. Percentage of trials that are successful for each animal with light on or off in each group (# successful trials during light on (OR off) / # of all trials with light on (OR off)) at late training (**C**) and early training (**I**); error bars, SEM; two-way ANOVA, multiple comparisons; p = 0.703, control; p = 0.0014, hChR2, p = 0.0148 for group x condition for (**C**), p = 0.977, control; p = 0.102, hChR2, p = 0.133 for group x condition for (**I**). Velocity traces (cm/s) of all trials at late training (**D**) and early training (**J**) from control, top,

and hChR2, bottom, animals during light, blue, and no light trials, gray; 0=trial start; blue bar denotes time of light, p > 0.999 (control), p < 1 × 10⁻¹⁵ (hChR2) for (**D**) and p > 0.999 (control), p = 0.000323 (hChR2) (**J**), time x condition, two way ANOVA; solid line, mean; shaded area, SEM. Maximum velocity (cm/s) of all trials for each animal with light on or off in each group at late training (**E**) and early training (**K**); two-way ANOVA, multiple comparisons, p = 0.871 (control), p = 0.0003 (hChR2), p = 0.0026 for group x condition for (**E**); p = 0.981 (control), p = 0.0331 (hChR2), p = 0.0642 for group x condition for (**K**); Wheel distance (cm) traveled during all trials for each animal with light on or off for each group late training (**F**) and early training (**L**); two way ANOVA, multiple comparisons, p = 0.951, (control), p = 0.0027, hChR2, p = 0.0139 for group x condition for (**F**), p = 0.592 (control), p = 0.191 (hChR2), p = 0.0837 for group x condition for (**L**). See Supplementary Table 3 for summary of statistics.

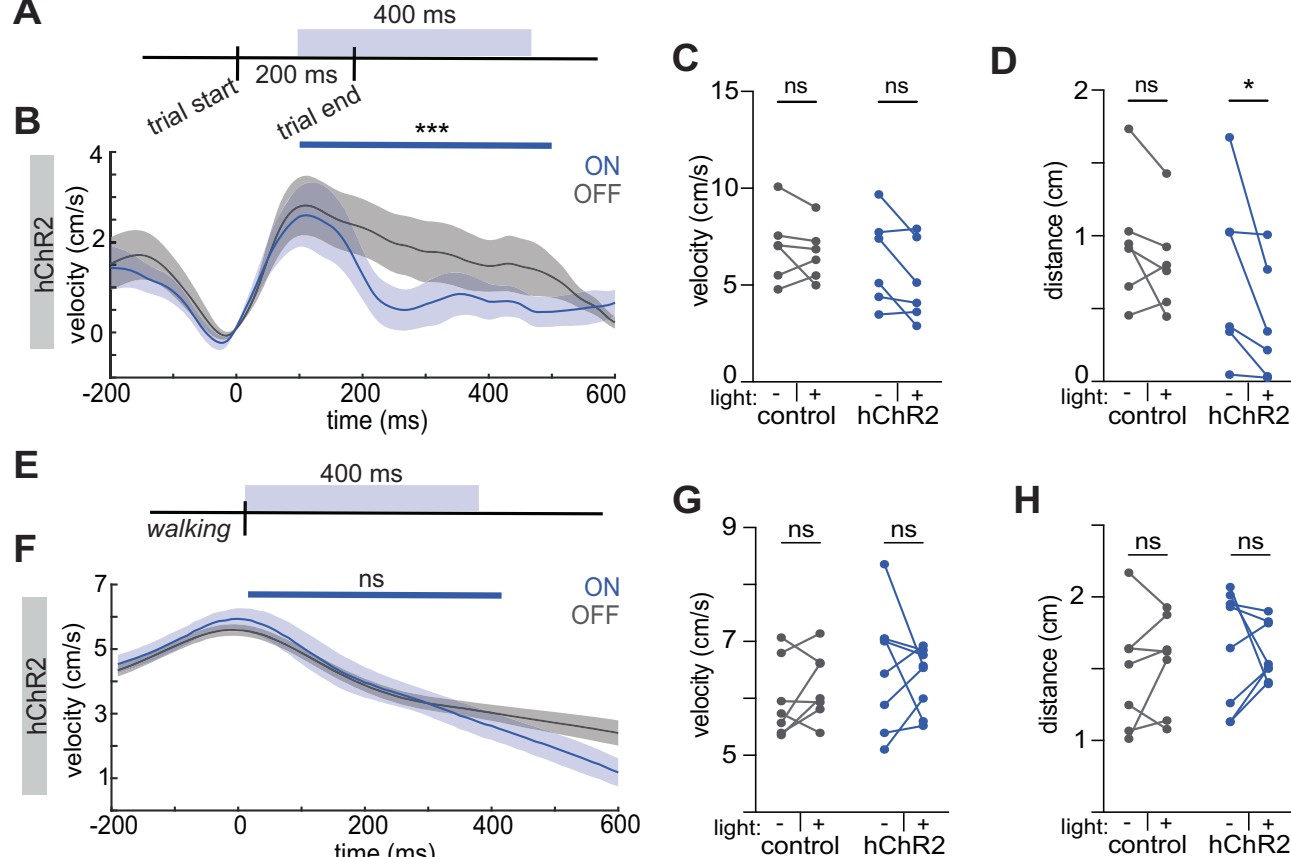

**Fig. 4 | Differential perturbation to movement during manipulations of M1^CT activity.** Schematic of optogenetic light delivery for closed loop at velocity max (**A**) and closed loop during walking bouts (**E**). Velocity traces (cm/s) of all trials from hChR2 animals (see Figs. S10A and S11A for controls) during light, blue, and no light trials, gray; 0 = trial start; blue bar denotes time of light; solid line, mean; shaded area, SEM; time x condition, two way ANOVA, for closed loop at velocity max, p = 0.0003 (**B**) and closed loop during walking bouts, p = 0.996 (**F**). For **B**–**D**: N = 6 animals for both groups; for **F**–**H**: N = 7 animals, control; N = 8 animals, hChR2. Maximum velocity (cm/s) of all trials for each animal with light on or off in each

group; two-way ANOVA, multiple comparisons, for closed loop at velocity max, p = 0.733, (control), p = 0.0789 (hChR2), p = 0.277 for group x condition (**C**) and closed loop during walking bouts, p = 0.698 (control), p = 0.765 (hChR2), p = 0.324 for group x condition (**G**). Distance (cm) traveled during all trials for each animal with light on or off for each group; two-way ANOVA, multiple comparisons, for closed loop at velocity max, p = 0.513 (control), p = 0.0406 (hChR2), p = 0.267 for group x condition (**D**) and closed loop during walking bouts, p = 0.800 (control), p = 0.961 (hChR2), p = 0.546 for group x condition (**H**). See Supplementary Table 3 for summary of statistics.

measures ANOVA, see Supplementary Table 3 for summary of all statistics; N = 11 mice, control, N = 11 mice, hChR2). These observations suggest that perturbing the activity of these neurons affects learned movement regardless of the phase during movement execution.

We next investigated if the activity of M1^CT neurons had the same suppression during a more naturalistic movement that does not require extensive training. We decided to focus on locomotion because a) it is a movement that engages the forelimb in a different

biomechanical sequence as well as many other muscle groups to generate a whole-body movement, and b) circuits in M1 as well as downstream circuits in the midbrain and hindbrain that mediate locomotion are distinct from those that are engaged in task specific forelimb movement[12,50] providing an interesting comparison for our forelimb reaching task. We utilized the same two-photon imaging approach as before to record calcium dynamics in the dendrites of FoxP2 + M1^CT neurons. The predominant population still consisted of

downmodulated neurons as in the wheel pulling task although at a somewhat lower proportion (Fig. S13A, B). To examine if the suppression of M1$^{CT}$ activity is critical for walking, we performed similar closed-loop optogenetic perturbations of FoxP2-cre animals expressing either a control fluorescent protein or channelrhodopsin (hChR2) as head fixed mice performed self-initiated walking bouts on a running wheel. Light was delivered after mice had commenced a walking bout (20 hz; 10 ms pulse width; Fig. 4E). Here, we observed no difference in how mice walked when comparing light on or off trials in either the opsin or control group (Figs. 4F and S13C, D; for wheel velocity: p = 0.996 and p > 0.999 for opsin and control, respectively; two-way ANOVA for 400 ms after light on, time x condition; for limb movement: p = 0.992 and p = 0.848 for control and hChR2, respectively; see Supplementary Table 3 for summary of all other statistics; N = 11 mice, control, N = 11 mice, hChR2), the max velocity reached during the light delivery period (Fig. 4G, p = 0.698 and p = 0.765 for control and hChR2, respectively, two-way repeated measures ANOVA; see Supplementary Table 3 for summary of all other statistics, N = 6 both groups), or distance traveled (Fig. 4H, p = 0.800 and p = 0.961 for control and hChR2, respectively, two-way repeated measures ANOVA; see Supplementary Table 3 for summary of all other statistics, N = 7 mice, control, N = 8 mice, hChR2).

### M1$^{CT}$ regulation of M1 corticospinals scales with learning

Our findings that M1$^{CT}$ neurons are suppressed during the execution of learned forelimb movements contrast with the positive correlation with movement reported for other output populations of motor cortex, notably corticospinal neurons of Layer V[27]. We confirmed the increased activity of M1 corticospinal neurons during our wheel turning task as well as locomotion by using a retrograde labeling approach from the cervical spinal cord (Fig. S14A) and imaging the dendritic dynamics of this population through a cranial window. In both wheel turning and walking, the activity of this population was positively correlated with movement (Figs. 5A, B and S14B), and the predominant population consisted of up modulated neurons (Fig. S14C, D). Remarkably, we observed that the magnitude of this correlation with movement increased during training (Figs. 5C and S14E, F), a pattern symmetric to what we observed for M1$^{CT}$ neurons.

Given the opposing activity of these two populations, we hypothesized that M1$^{CT}$ neurons could drive feedforward inhibition of corticospinal neurons in a similar circuit configuration to that reported in other cortical regions such as visual cortex[51,52] and somatosensory cortex[53,54]. In this model, suppression of M1$^{CT}$ neurons would release feedforward inhibition and allow for increased corticospinal activity. To address this possibility, we performed whole cell voltage-clamp recordings from fluorescent, retrogradely labeled corticospinal neurons in acute, live brain slices expressing Cre-dependent hChR2 in M1$^{CT}$ neurons (Figs. 5D and S15A, B). Here, we utilized the Ntsr1-cre mouse to avoid any hChR2 expression in Layer V. We separately confirmed that photostimulation (100 ms) drove direct excitatory currents in M1$^{CT}$ neurons (Fig. 5E), as well as the baseline firing properties of M1$^{CT}$ neurons (Fig. S15C–E). We found hChR2 stimulation of M1$^{CT}$ evoked relatively small excitatory postsynaptic currents (EPSCs) and inhibitory postsynaptic currents (IPSCs) in M1 corticospinal neurons when recording at membrane holding potential of −70 mV or 0 mV, respectively (Fig. 5F, top, G, H, J, gray). Excitatory and inhibitory responses were rapidly induced within 7 ms and 17 ms of stimulus onset respectively (Fig. S15G, I, K). These findings are consistent with prior reports of feedforward inhibition from M1$^{CT}$ to corticospinal neurons[40]. Bath application of GABAzine eliminated IPSCs, confirming these inhibitory photocurrents are mediated by GABA$_A$ receptors (Fig. S15F, p = 0.0284, paired t-test). Interestingly, we observed that the magnitude of this inhibition changed with training (Fig. 5F, G, J) with a significant increase in the peak inhibitory current (Fig. 5H, p < 0.001, untrained inhibitory peak currents, two-way ANOVA, see

Supplementary Table 3 for summary of statistics; n = 22 neurons untrained, n = 24 neurons trained; N = 2). This change corresponded with a higher inhibitory to excitatory ratio in corticospinal neurons with training, as the change in excitatory responses was less pronounced (Fig. 5G, p = 0.0136, two-tailed t-test). The latency of these responses, however, remained similar to those of untrained mice. Excitatory and inhibitory responses were detected within 6 ms and 13 ms of stimulus onset respectively (Fig. S15H, J, L, M). Together, this led us to conclude that M1$^{CT}$ neurons regulate the activity of corticospinal neurons through feedforward inhibition, whereby suppression of M1$^{CT}$ neurons would result in disinhibition of corticospinal neurons. Finally, we observed that this feedforward inhibition was strongly facilitated with learning (Fig. 5H).

## Discussion

Our studies identified M1$^{CT}$ neurons as a key permissive population for motor execution. We first identified this population using an activity-based, single-cell RNA-seq based screen of M1 during the progression of a motor task. Experiments characterizing the dynamics of M1$^{CT}$ neurons showed that most of these neurons have a marked suppression during movement, and that this suppression is more substantial in late training than early training. Accordingly, closed-loop experiments revealed that optogenetic activation of these neurons during late training at different phases of movement execution rapidly and substantially impaired execution, suggesting that the decrease in activity is permissive for movement execution. However, closed-loop optogenetic activation of M1$^{CT}$ neurons early in training or during locomotion had less impact on movement. Finally, we demonstrate that activity of M1$^{CT}$ neurons suppresses the activity of M1 corticospinal neurons through feedforward inhibition, and that the strength of inhibition changes with learning. Our experiments uncover a mechanism by which suppression of activity of M1$^{CT}$ neurons can disinhibit corticospinal neurons permitting them to fire during learned forelimb movements.

Our activity-based screen fills a niche in exploring cell type specific contributions to short time scale behaviors. Identifying the cell types with patterns of engagement during a particular behavior complements hypothesis-driven investigation of the role of particular cell types. This can then be followed by confirmatory techniques for neuronal activity recording, such as calcium imaging, which allow for in-depth characterization of particular activity patterns. Whereas we chose to focus on a cell type that differed in enrichment with training, it may also be informative to examine cell types that constitute the largest part of the active ensemble at a given point. In our experiments, we observed greater numbers of several IT subsets at the time points assessed. This could imply that these neurons are generally active for cortical processing regardless of the specific ongoing behavior. Although our screen has limitations and is not comprehensive, it still allowed us to compare differential cell type engagement during training. We also chose to deliver light unbiasedly through the training session to include all activity that could contribute to learning, not just movement execution, but groups interested in more specific features of a given behavior could refine periods of light delivery. As new tools emerge, they may allow for similar but more quantitative approaches for identifying active cell types in the future.

There has been a long-standing interest in defining the role of the primary motor cortex and its relationship to movement. Prior studies have highlighted the heterogenous patterns of activity relative to movement, but narrowing these examinations to specific cell types has begun to produce a more precise relationship between neural activity and movement. For example, we and others have noted that most M1 corticospinal neurons, which project directly to the spinal cord and are hence relatively proximal to skeletal muscle, have predominantly movement correlated activity[27]. Similarly, intratelencephalic neurons projecting within cortex show increased

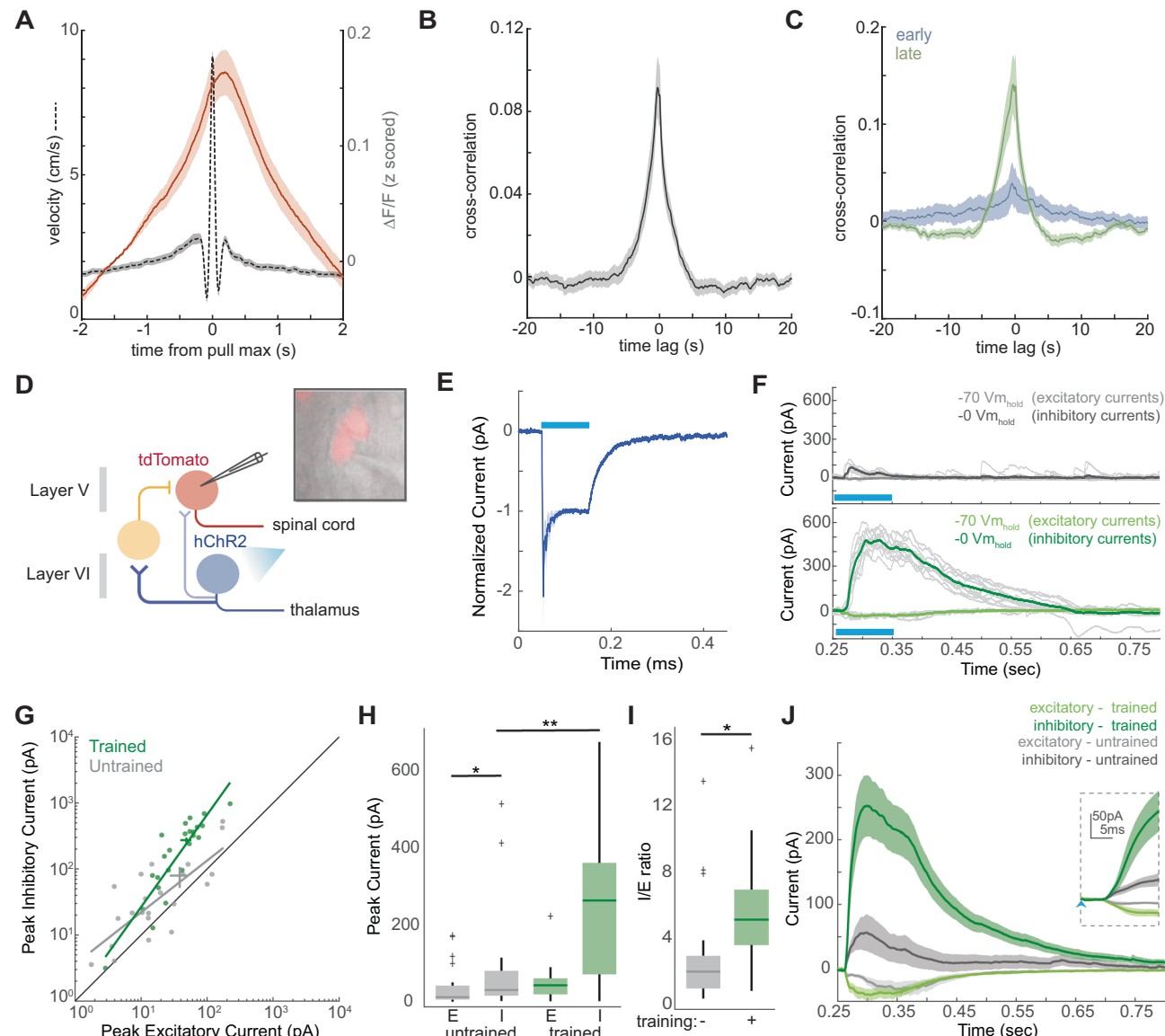

**Fig. 5 | Multisynaptic inhibition of corticospinal neurons by M1^CT neurons augments with training. A** Z-scored ΔF/F during wheel pulls; mean, solid red line; shaded area, SEM; Wheel velocity (cm/s) during same trials, dotted line; **B** Cross-correlation of Z-scored calcium ΔF/F to wheel velocity for whole session; mean, solid gray line; shaded area, SEM; n = 66 units from N = 3 animals for (**A**) and (**B**). **C** Cross-correlation to wheel velocity for pulls at early (blue) and late (green) training sessions; solid line, mean; shaded area, SEM. n = 20 units, early; n = 22 units, late; from N = 3 animals. **D** Schematic of recording strategy from tdTomato labeled corticospinal neurons while activating M1^CT neurons expressing hChR; putative inhibitory neuron, yellow; Created in BioRender. Carmona (2025) https://BioRender.com/j64k839. **E** Population grand average response of corticothalamic neurons to photostimulation; light, blue bar; solid line, mean; shaded area, SEM; responses are normalized to the steady-state evoked current. **F** Recordings from exemplar corticospinal neurons after 100 ms photostimulation (blue bar) from untrained, top, gray, or trained, bottom, green, animals at either −70 mV holding potential, to measure excitatory currents, light gray and light green, or 0 mV

holding potential, to measure inhibitory currents, dark gray or dark green; thick line, mean; thin lines, single trials. **G** Pairwise comparisons of peak excitatory and inhibitory currents from corticospinal neurons from untrained, gray, or trained, green, animals. Green and gray line, first polynomial linear fit, $R^2 = 0.730$, untrained, $R^2 = 0.760$, trained, simple linear regression. Comparison of peak excitatory (E) or inhibitory (I) currents (**H**) or inhibitory to excitatory ratio (**I**) from untrained, gray, or trained, green, animals; center line, mean; range method, Tukey. For **H**, $p = 7.47 \times 10^{-5}$, peak inhibitory current in trained v untrained and $p = 2.28 \times 10^{-6}$ for trained peak inhibitory current v trained peak excitatory current, two-way ANOVA, multiple comparisons; for **I**, $p = 0.0136$, two-tailed unpaired t-test; see Supplementary Table 3 for summary of statistics. **J** Grand average excitatory responses, light, or inhibitory responses, dark, from all neurons; mean, solid line; shaded area, SEM; inset for first 18 ms following the onset of photostimulation (indicated with blue arrow); for **G**–**J**, n = 22 neurons from N = 2 mice, untrained and n = 24 neurons from N = 2, trained.

activity during movement[31]. In contrast, our imaging of M1^CT neurons shows that the majority of this population reduces activity at the time of movement. This signature has also been recently reported for corticopontine neurons of layer V[31] highlighting the heterogeneity within M1 layers. While these examinations allow us to begin integrating the roles of these cell types during movement, it remains to be seen whether much more refined targeting of smaller subsets of

neurons as defined by transcriptomic profiling will yield a more complete picture of the role of M1.

Our closed-loop optogenetic experiments suggest that activation of M1^CT neurons at a time when they should be suppressed can rapidly and profoundly impair movement execution. Our experiments further suggest that feedforward inhibition from M1^CT neurons to corticospinal neurons, (Fig. 5) which are positively correlated with movement

and are thought to relay motor commands throughout the brain and spinal cord. The change we observed in the magnitude of inhibition with training is also reflected in the magnitude of movement perturbation observed in our optogenetic experiments at early or late training. This further highlights this mechanism as a potential source of refinement that integrates the activity of these population during learning. In addition, our activity-based labeling screen demonstrated an increase in the activity of M1$^{CT}$ neuron with training. Although they are silent during movement execution, their activity may be critical in suppressing or regulating activity of other movement generating populations either to help in movement selection or to refine the timing of movement execution.

It should also be noted that among the targets of corticospinals are many of the same thalamic regions as those targeted by layer VI corticothalamic neurons. We demonstrated that within layer V and VI, similar patterns of topographical organization occur withing M1 relative to thalamic targets (Fig. S3) indicative of a columnar organization. This coupling of output targets underscores the relevance of local M1 mechanisms to regulate output signals from M1. In the future, it would be interesting to examine the activity of these thalamic targets regions and to investigate any additional mechanism for integrating these signals within both M1 and thalamus. For example, additional modulation may be provided by M1$^{CT}$ neurons to corticospinal neurons through connections with the reticular nucleus and signals back to cortex through thalamocortical neurons. Furthermore, our characterization of corticothalamic neurons focused on those within the caudal forelimb area, but our projection mapping indicates topographical specificity that extends to other regions of motor cortex (Fig. S3). It will also be important to determine whether there is specificity in the activity or functional contribution of corticothalamic neurons relative to their location in motor cortex and their thalamic targets.

It is intriguing to note that a similar mechanism of local, disynaptic inhibition by layer VI neurons to all other layers has been described in visual and somatosensory cortices[51–54]. Our parallel observation in motor cortex brings up avenues of further exploration. For example, it would be informative to examine whether M1$^{CT}$ neurons also drive a similar inhibitory input to more superficial layers within motor cortex, such as layer II/III which is thought to perform a more distinct role in processing inputs to M1 rather than relaying output signal as do deeper layers. Likewise, it will be interesting to explore more general functions of this modulation in M1. In visual cortex, the inhibitory modulation provided by layer VI neurons to superficial layers plays an important role in gain control. We observed a gradation in the suppression of M1$^{CT}$ neurons relative to movement vigor (Fig. 2H–K), suggesting a similar modulatory role for movement. If true, this could suggest a common role for this population and this circuit motif across different regions of cortex.

## Limitations of the study

Our use of CaMPARI, a calcium and light-dependent indicator of neuronal activity, allowed us to assess the composition of neurons with high calcium entrance during different sessions across training. A caveat with this approach, however, is that light delivery through a cranial window in our experiments provides higher power light to more superficial neurons than those in deeper layers of cortex. We have controlled for this by only comparing each subpopulation to itself across training timepoints. This has precluded comparisons between subpopulations with regards to their relative contribution to the active neuronal population at each of the timepoints because of this additional variable. It is also possible that for populations that extend across various layers, that their enrichment could be underrepresented if neurons in deeper locations are predominantly engaged. Similarly, an approach where we performed our screen when animals achieved a particular proficiency metric could have decreased potential variability between animals. This was not possible in our approach as we chose to utilize single-cell RNAseq with CaMPARI, which is cytosolic, preventing the use of single-nuclei RNAseq that allows for freezing of samples over multiple days. Thus, to obtain sufficient numbers of cells, multiple animals had to be pooled on the same day. As new tools are developed, it will be possible to assess individual animals and provide performance aligned molecular characterization. It should also be noted that light for CaMPARI tagging was delivered unbiasedly throughout the session and is hence not a measure solely of activity during movement. Finally, it is also possible that some CaMPARI labeling was a result of sub-threshold synaptic inputs[55], and the results of our screen should be interpreted to include neurons generally engaged, both receiving inputs and relaying outputs, during the labeling window.

With regard to our calcium imaging, we have imaged dendrites to allow for imaging of deep populations without an additional invasive implant. All data presented throughout is not somatic and is not directly spikes but the extracted calcium transients from our two-photon imaging. Our optogenetic manipulations also consisted of the same frequency of pulsed light which does not replicate the most physiological conditions. Finally, we would like to highlight while we have compared a more skilled task (wheel pulling) and a less skilled task (locomotion), it remains to be determined whether a specific aspect of the wheel pull (i.e. reach, grasp, or pull) requires the decrease in activity in M1$^{CT}$ neurons. Tasks that isolate each movement component could help to further examine this feature in future experiments.

## Methods

### Mice

All experiments and procedures were performed according to National Institutes of Health (NIH) guidelines and approved by the Institutional Animal Care and Use Committee of Columbia University. Adult mice of both sexes, aged 2–6 months, were used for all experiments, except for RNA sequencing which only used males aged 3 months. The strains used were: C57BL6/J (Jackson Laboratories, 000664), FoxP2-Cre (Jackson Laboratories, 030541), and Ntsr1-Cre (Mutant Mouse Resource & Research Centers, 030648-UCD). Only mice used for RNA sequencing experiments were individually housed. All mice were kept under a 12-h light-dark cycle. All mice except those used for calcium imaging experiments were kept under reverse light cycle conditions.

### Viral vectors

For CaMPARI based experiments, AAV1-hSyn-CaMPARI (Addgene 100832-AAV1) or AAV1-hSyn- FLEX-CaMPARI (custom prep; Janelia viral core) were used. Projection mapping was performed with the membrane bound GFP expressed from AAV8-hSyn-JAWS-KGC-GFP-ER2 (Addgene 65014-AAV8). Calcium imaging experiments were performed with AAV1-Syn-FLEX-jGCaMP7f (Addgene 104492-AAV1) for M1$^{CT}$ neurons or AAVretro-hSyn-FLEX-jGCaMP7f (Addgene 104492-AAVrg) in combination with AAV1-hSyn-Cre (Addgene 105553-AAV1) for corticospinal neurons. Optogenetic experiments used AAV1-Ef1a-DIO-EYFP (Addgene 27056-AAV1) and AAV1-EF1a-DIO-hChR2 (H134R)-EYFP (Addgene 20298-AAV1) for control and opsin animals respectively. Slice electrophysiology experiments used AAV1-EF1a-DIO-hChR2 (H134R)-EYFP (Addgene 20298-AAV1) injected in M1 and AAVretro-CAG-TdTomato (Addgene 59462-AAVrg) injected into spinal cord for photostimulation experiments. Baseline recordings used AAV5-hSyn-DIO-mCherry (Addgene 50459-AA5).

### Stereotactic surgery

Animals were anesthetized using isoflurane, and analgesia was delivered subcutaneously in the form of either carprofen (5 mg/kg) or burprenorphine SR (0.5-1 mg/kg) as well as bupivacaine (2 mg/kg). For all viral vectors delivered to M1, injections were centered in the caudal forelimb area (CFA) using the following coordinates: 1.5 mm lateral to the midline, 0.25 mm rostral to bregma. A total of 5 injections were made in this region to cover a large portion of CFA in the contralateral (left) hemisphere. One injection was at the center of these coordinates,

with 4 additional injections either 300 um anterior/posterior to this or 300 um medial/lateral to this. Specific sites were adjusted to avoid blood vessels. A Nanoject III Programmable Nanoliter Injector (Drummond Scientific) was used at a rate of 1 nl/s.

For the CaMPARI screen experiments, a total of 1 uL of virus was delivered (200 nL at each of the five injection sites) at a range extending to 800 um below the pial surface (25 nL every 100 um). For closed-loop labeling with CaMPARI, a total of 300 nL of virus was delivered (75 nL at each of the four injection sites) at a range extending from 800–600 um below the pial surface (15 nL every 50 um). For projection mapping, a total of 500 nL of viral vector was injected (100 nL per injection site) from 900–500 um below the pial surface (20 nL at every 100 um). For calcium imaging of M1$^{CT}$ neurons, a total of 375 nL of viral vector was injected (75 nL per injection site) from 950–650 um below the pial surface (25 nL at every 100 um). In FoxP2-cre mice or for imaging corticospinal neurons, a total of 1 uL was injected into the right cervical spinal cord (200 nL per segment from C3-C7). For optogenetic experiments, a total of 500 nL of viral vector was injected (100 nL per injection site) from 850–650 um below the pial surface (20 nL at every 100 um). For photostimulation slice recordings, a total of 240 nL was delivered (60 nL at each of 4 injection sites) at a range extending from 650–850 um below the pial surface (20 nL every 100 um) along an injection in the right cervical spinal cord of a total of 1 uL (200 nL per segment from C3-C7). For baseline recordings, a single injection of 50 nL of diluted virus ($2.1 \times 10^{12}$ GC/mL) was delivered at 800 um below the pial surface. To prevent backflow, we waited 3-5 minutes before the initial injection and 5-10 minutes after each injection. For animals used in behavioral experiments, a headplate was attached to the skull with Metabond (Parkell) post injection.

For retrograde labeling from thalamus, we used the following coordinates, for the ventral region: 1.3 mm lateral from the midline, 1.22 mm caudal to bregma, and 3.5 mm below the pial surface; for the lateral region: 1.25 mm lateral from the midline, 2.18 mm caudal to bregma, and 3.25 mm below the pial surface; for the medial region: 0.1 mm lateral from the midline, 1.46 mm caudal to bregma, and 4.2 mm below the pial surface. A total of 10 nL of 4% FluoroGold (Fluorochrome) was injected. FluoroGold was freshly diluted the day of injection from a 10% stock solution. To prevent backflow, we waited 2 min before each injection and 10 min after each injection.

## Cranial window implantation

For animals requiring a cranial window, surgery was performed as described above with the following modification. Animals were anesthetized using isofluorane, and analgesia was delivered subcutaneously in the form of burprenorphine SR (0.5-1 mg/kg) and bupivacaine (2 mg/kg). Dexamethasone (2 mg/kg) was also delivered as an anti-inflammatory agent. A larger craniotomy was made (2.5 mm in diameter) for a custom cranial window made of a glass plug (2.5 mm diameter) attached to a larger glass base (3.5 mm in diameter) with optical cement (Norland Optical Adhesive 61). After viral injection, the cranial window was implanted to gently press on the brain at the site of the craniotomy and secured with Metabond (Parkell). Headplate implantation followed cranial window implantations.

## Behavior

For the wheel pulling task, several noise attenuating behavioral chambers equipped with IR light were assembled with custom components to train several animals in parallel. Mice were placed in flat bottom, opaque tube, and head fixed using a custom metal headplate by screwing in place to adjustable head posts. The small rotary wheel was directly below the right forelimb of the animal when in the holder. A small partition prevented animals from utilizing their left forelimb to pull the wheel. A small screw was also provided for left forelimb placement. Mice were habituated to head fixation and water delivery via

the water port for several days before the beginning of training. The water port consisted of a blunt needle positioned in front of the mouse so water was reachable by licking. Water droplets were generated using a solenoid valve and were calibrated before the beginning of training for each experiment. The 60 mm diameter rotary wheel was assembled from two acrylic sides held together by small metal screw rungs. An absolute rotary encoder (US Digital; 10 bit) was placed in the shaft of the wheel to assess wheel movement. Each rung was wired to a capacitance sensor. An IR beam was placed at the apex of the wheel on a custom 3D printed holder allowing for alignment of the beam. Data from all behavioral components was collected using a DAQ (National Instruments) sampling at 2000 Hz. Behavioral sessions were controlled using custom MATLAB code which assessed beam breaks and changes in rung capacitance. If thresholds were met for wheel rung capacitance, IR beam cross, and a minimum threshold velocity to exclude minor wheel jitter, a trial would commence. Wheel displacement was then assessed, smoothed, and converted to velocity. If the threshold velocity was surpassed for success, a pulse was sent to the solenoid to deliver the water reward. Mice were habituated to the task during three pre-training sessions prior to the beginning of training. During these sessions, mice were exposed to the task but at low trial velocity thresholds for a limited number of trials. Training sessions consisted of 30 min daily or a maximal of 100 successful trials. Mice were trained during the dark phase of their reversed light cycle housing schedule.

For locomotion, mice were headfixed atop a large acrylic rotary wheel places inside a noise attenuating chamber. The wheel was lined with clingwrap for traction. Mice were habituated to head fixation before the beginning of training. A quadrature rotary encoder (US Digital; 1024 CPR) was placed in the shaft of the wheel to assess wheel movement. The behavioral assays were controlled and data was collected using pyControl.

## CaMPARI labeling

For our activity screen, we allowed four weeks post injection for viral expression. Mice were then head-fixed as in all behavioral experiments and a plastic fiberoptic patch cord (960 um core; 0.63 NA) ending in a metal ferrule (Doric) was placed directly over the cranial window. A multichannel LED driver (Doric) was used with 405 nm central wavelength connectorized LED (Doric, ~60 mW at fiber tip or 82.89 mW/mm$^2$ at the cranial window. See table below for estimates of irradiance at various cortical depths with these experimental parameters). Concurrent with the start of the training session, light delivery was initiated using the Doric Neuroscience Studio software. Light was delivered in square wave pulses of 5 s each with 5 s of light off between each pulse for a total of 12 min. For closed loop labeling, light was delivered using the same setup, but instead a pulse was triggered at the start of a trial (for trial ON labeling) or outside of trial starts (for trial OFF labeling). Pulses consisted of a 500 ms square pulse with at least 2 seconds between pulses.

| Cortical depth (um) | Irradiance (mW/mm²) |
|---|---|
| 100 um | 49.282 |
| 200 um | 29.299 |
| 300 um | 17.419 |
| 400 um | 10.356 |
| 500 um | 6.157 |
| 600 um | 3.66 |
| 700 um | 2.176 |
| 800 um | 1.294 |
| 900 um | 0.769 |
| 1000 um | 0.457 |

Irradiance values calculated by the following:
Irradiance at the fiber tip ($E_o$):

$$E_0 = \frac{P}{\pi\left(\frac{d}{2}\right)^2} \quad (1)$$

$P$: Power = 60 mW = 0.06 W
$d$ : diameter of fiber = 960 uM = 0.00096 m

$$E_0 = \frac{0.06}{\pi\left(\frac{0.00096}{2}\right)^2}$$

$$E_0 = 82893 \, W/m^2$$

Irradiance at various cortical depths:

$$E(z) = E_0 \cdot e^{-\mu_t z} \quad (2)$$

$z$: depth
 $\mu_t$ : total attenuation coefficient
To calculate the total attenuation coefficient ($\mu_t$):

$$\mu_t = \mu_a + \mu_s \quad (3)$$

$\mu_a$ : absorption coefficient
 estimated to be 0.2 mm$^{-1}$ [56]
$\mu_s$ : scattering coefficient
 estimated to be between 2 mm$^{-1}$ [56]

$$\mu_t = 0.2 + 5$$

$$\mu_t = 5.2 \, mm^{-1}$$

Sample calculation for 100 um below the surface:

$$E(100um) = E_0 \cdot e^{-\mu_t z}$$

$$E(100um) = 82893 \cdot e^{-5200(0.0001)}$$

$$E(100um) = 82893 \cdot e^{-0.52}$$

$$E(100um) = 82893 \cdot 0.5945$$

$$E(100um) = 49,281.59$$

### Generation of single-cell suspensions and FACS

After CaMPARI labeling, mice were returned to their home cage for 15 min and then euthanatized by transcardial perfusion. Subsequent steps were adapted from previous studies describing a protocol optimized for preserving transcriptional state[57]. Perfusions were performed with with ice-cold choline solution: 2.1 g/l NaHCO3, 2.16 g/l glucose, 0.172 g/l, NaH2PO4 * H2O, 7.5 mM MgCl2•6H2O, 2.5 mM KCl, 10 mM HEPES, 15.36 g/l choline chloride, 2.3 g/l ascorbic acid, and 0.34 g/l pyruvic acid. Thick sections were cut using a matrix, and M1 was quickly dissected. The dissected M1 tissue was further triturated with spring scissors. The tissue was then digested with the Papain dissociation system (Worthington) according to the manufacturer's instructions. However, the EBSS solution was replaced with the following: HBSS (Life Technologies), 10 mM HEPES (Sigma), 172 mg/l kynurenic acid (Sigma), 0.86 g/l MgCl2•6H2O (Sigma), 6.3 g/l

D-glucose (Sigma). Incubation with papain was conducted using a dialysis membrane (Fisher Scientific, Slide-a-Lyzer MINI Dialysis Device,10k molecular weight cut-off) suspended in a beaker of dissociation solution allowing for constant oxygenation (95% $O_2$ and 5% $CO_2$) of the solution without disrupting enzyme integrity within the dialysis membrane. After enzymatic dissociation, the tissue was broken into a single-cell suspension by gently pipetting using wide-bore pipette tips. The sample was cleaned using the Debris Removal Solution (Miltenyi Biotec) as described by the manufacturer. Cells were suspended in dissociation solution containing 0.04% BSA, Vybrant DyeCycle Ruby for exclusion debris, and RNAase inhibitors. Cell sorting was performed using a FACSAria Cell Sorter (Becton Dickinson) with a 130 um nozzle at 12 PSI using the following laser lines and filters: CaMPARI Green, 488 nm, 530/30 bandpass; CaMPARI Red, 561 nm, 610/20 bandpass; DyeCycle Ruby, 637 nm, 670/30 bandpass. Gates were set using a sample not expressing CaMPARI and a sample not exposed to photoconversion light. Any cells with detectable red fluorescence were collected.

### Single-cell RNA-seq

Sorted cells were captured and barcoded using 10x Genomics Chromium v3 according to the manufacturer's protocol. Samples were processed and libraries were prepared and sequenced by the JP Sulzberger Columbia Genome Center Single Cell Analysis Core.

### Single-cell RNA-seq analysis

Single-cell RNA-seq analysis followed a standard workflow, comprising the following steps:

(1) Sequencing read files were aligned to the mm10 genome using 10x Genomics Cellranger v.3.1.0, generating a table of cell barcodes by Unique Molecular Identifier (UMI) counts per gene.

(2) Raw counts files were aggregated, and cells with >20% UMIs mapping to mitochondrial genes were removed, as well as cells with <200 UMIs after removing mitochondrial genes, ribosomal protein genes, pseudogenes, and gene models.

(3) Cells were clustered using the Seurat R package (v.4.0.0), with SCTransform (default parameters) and PCA, followed by integration across batches using the Harmony R package (with 40 PCs). The cell-cell neighbor network was constructed with 30 Harmony dimensions, and the Louvain community detection algorithm was run with resolution = 0.4.

(4) Based on expression of marker genes for neurons and glia (Snap25, Slc17a6, Slc17a7, Gad1, Gad2, Mbp, Mog, Fgfr3, Aqp4, Pdgfra, Tmem119, Aif1, Ptprc, and Cldn5), the clusters corresponding to glutamatergic and GABAergic neurons were extracted.

(5) Step 3 was re-run on the glutamatergic and GABAergic neurons (separately), with 30 PCs, 20 Harmony dimensions, and resolution = 0.2.

(6) Combinatorial marker genes were identified for each cluster using the FindMarkers command in Seurat, run on all pairs of clusters. In parallel, putative layer identities for each cluster were assigned by mapping to Azimuth Mouse Motor Cortex reference.

(7) To assess differential proportions among conditions, cell count values were run through the ANCOM-BC and MASC R packages, with pairwise comparison between control, early, and late conditions. All q-values were then further FDR (Benjamini-Hochberg) corrected.

### Projection mapping

Mice were euthanized by intracardial perfusion with 1× PBS followed by 4% paraformaldehyde. Brains were post-fixed in 4% paraformaldehyde overnight at 4 °C. A step-by-step protocol of the AdipoClear protocol is available[58] and this protocol was performed as described

with the following modifications. Primary and secondary antibody incubations were conducted at 37 °C and for 6 days per incubation. The following primary antibodies were used at a dilution of 1:2500: chicken GFP antibody (Aves: GFP1020; polyclonal) and rabbit RFP antibody (Rockland: 600-401-379; polyclonal). The following secondary antibodies were used at a dilution of 1:2500: donkey anti-rabbit Alexa Fluor 647 (ThermoFisher Scientific: A31573; polyclonal) and donkey anti-chicken Alexa Fluor 647 (Thermo Fisher Scientific: A78952; polyclonal). Samples were imaged with a light-sheet microscope (Ultramicroscope II, LaVision Biotec) using a 4x objective for the fluorescent proteins or 1.3x objective for autofluorescence imaging. The ClearMap pipeline was used to map the data to the Allen Brain Reference Atlas[59]. All signal mapped to thalamus was summed and the percent of signal in each nucleus was calculated for each animal.

### Retrograde mapping
Two weeks post injection, mice were euthanized by intracardial perfusion with 1× PBS followed by 4% paraformaldehyde. Brains were post-fixed in 4% paraformaldehyde overnight at 4 °C. Serial coronal sections (50um) were collected and stained with NeuroTrace 640/660 (Thermo Fisher Scientific). Imaging was performed with an AZ100 automated slide scanning microscope using a 4x objective (Nikon, 0.4 NA). Image processing, registration to the Allen Brain Reference Atlas, and automated counting were performed using BrainJ (https://github.com/lahammond/BrainJ)[60]. Cell counts per animal were summed over anatomical regions or binned according to their mapped position for comparative analysis.

### Two-photon imaging and analysis
All behavioral training and habituation were conducted as described above. Training did not commence until at least 8 weeks post viral injection. A behavioral assembly similar to those described for behavior was placed on a modified 2p microscope (Bruker) for calcium imaging experiments. A 25x water immersion objective (Olympus,1 NA) was used with a mode-locked Ti:sapphire laser (Verdi 18 W, Coherent) at 920 nm. Images were collected with Prairie View software (Bruker) at 64 Hz and averaged every 4 images for an effective sampling rate of 16 Hz. The pial surface of the brain surface was identified at the beginning of each imaging session, and fields of view were selected at least 300 um below the pial surface. Voltage recordings of the encoder and solenoid were also collected with Prairie View. The behavioral code was run from a separate computer, and all behavioral components recorded. The encoder signal was used to align the imaging to the behavioral recordings. Motion correction and signal extraction was performed using CNMF[48] with an autoregressive process $p$ of 2. For analysis of the whole session, the ΔF/F or deconvolved events were z-scored over the whole session for individual neurons and then aggregated across animals for further analysis. For analysis of trials or pulls, the ΔF/F or deconvolved events were z-scored to signal 2-1.5 s prior to the movement event for each neuron. Classification was performed by calculating the approximate integral using the trapezoidal method for 100 ms bins starting 2 s before pull max and extending 2 s after and z-scored to the first five bins (−2 to −1.5 from pull max). Neurons with at least two bins of the 3 proceeding and the 3 preceding the pull max (total of 600 ms) with a z-score greater than 2 or −2 were classified as up or down respectively. All other neurons were classified as not modulated. Analysis was performed using MATLAB 2021b.

### Optogenetic manipulations
All behavioral training and habituation were conducted as described above. On the days of optogenetic manipulations, a plastic fiberoptic patch cord (960 um core; 0.63 NA) ending in a metal ferrule (Doric) was placed directly over the cranial window. A multichannel LED driver (Doric) was used with 465 nm central wavelength connectorized LED (Doric; ~35 mW). For closed-loop experiments, a pulse was sent at trial start on one third of trials to the driver to trigger light delivery as programmed in Doric Neuroscience Studio software. Light was delivered for 400 ms at 20hz with a pulse width of 10 ms. For open-loop experiments, light delivery was triggered using a random number generator to give light for 10% of the session. Light was delivered for 1.2 s at 20 hz.

Data displaying group comparisons was averaged over the days of manipulations and then over the animals of each group. Metrics presented by animal, such as peak velocity and distance, was measured per trial and then averaged for each day of manipulations and then by animal. Analysis was performed using MATLAB 2021b and GraphPad Prism9.

Right limb position during wheel pull trials and during locomotion was extracted using DeepLabCut[61]. Frames were annotated to track the wrist. Euclidian distance was calculated to each subsequent frame within the trial. For wheel pull trials, the max was taken for each trial and averaged for each animal for either light on or light off trials. For locomotion, the Euclidian distance from frame to frame was summed. All trials were averaged for each animal for either light on or light off trials.

### Slice electrophysiology
Mice were deeply anesthetized with isoflurane and transcardially perfused with an ice-cold carbogenated solution of HEPES-sucrose artificial cerebrospinal fluid (ACSF) containing 110 mM NaCl, 10 mM HEPES, 25 mM glucose, 75 mM sucrose, 7.5 mM MgCl$_2$, and 2.5 mM KCl. The brain was removed from the skull and glued to the stage of a vibrating microtome (Leica). Next, 300-μm coronal brain slices were cut in a bath of ice-cold, slushy, HEPES-sucrose ACSF. Slices were incubated for 30 min in a 34 °C bath of normal carbogenated ACSF containing 124 mM NaCl, 2.7 mM KCl, 2 mM CaCl$_2$, 1.3 mM MgSO$_4$, 26 mM NaHCO$_3$, 1.25 mM NaH$_2$PO$_4$, 18 mM glucose and 0.79 mM sodium ascorbate. Slices were then transitioned to room temperature, where they remained for the duration of the experiment. Patch electrodes (1–3 MΩ) were filled with a cesium/QX-314-based internal solution containing 5 mM QX-314, 2 mM ATP magnesium salt, 0.3 mM GTP sodium salt, 10 mM phosphocreatine, 0.2 mM EGTA, 2 mM MgCl$_2$, 5 mM NaCl, 10 mM HEPES, 120 mM cesium methanesulfonate and 0.15% Neurobiotin. All recordings were made using a Multiclamp 700B amplifier, the output of which was digitized at 10 kHz (Digidata 1440 A). Series resistance was always <20 MΩ and was compensated up to 90%. Neurons were targeted with DIC microscopy and epifluorescence when appropriate. Excitatory or inhibitory currents were isolated by clamping membrane voltage at −70mV or 0 mV, respectively. GABA$_A$ currents were blocked by superfusion of 10uM Gabazine for 10 min. Neurobiotin-filled cells were visualized post hoc through streptavidin processing. Brain slices were fixed in 4% paraformaldehyde for 1 h and rinsed several times in PBS. Slices were initially permeabilized in 1% Triton-containing PBS for 1 h at room temperature. Slices were incubated overnight at 4 C in 0.3% Triton PBS containing 0.1% Streptavidin DyLight Fluor 405, followed by several PBS rinses at room temperature.

### Histology and imaging
Images of CaMPARI photoconversion were obtained from mice trained as described above. Images of GCaMP expression were obtained from mice used for imaging once all training was complete. All mice were euthanized by intracardial perfusion with 1× PBS followed by 4% paraformaldehyde. Brains were post-fixed in 4% paraformaldehyde overnight at 4 °C. Coronal sections (50um) were cut with a vibratome. DAPI was used as a counterstain. GCaMP signal and hChR2-EYFP expression was amplified with chicken GFP antibody (Aves). TdTomato expression was amplified with rabbit RFP antibody (Rockland). For CaMPARI slices, imaging was performed with a confocal microscope (Zeiss 880) with a 20x objective within a week of mounting. All other images were acquired with a W1-Yokogawa spinning disk confocal microscope with a 4x objective for zoomed out panels or 20x objective for zoomed in panels.

## Statistical analysis

Statistical parameters, statistical tests, and statistical significance are reported throughout. Significance is defined as P < 0.05 with significance annotations of *P < 0.05, **P < 0.01, ***P < 0.001 and ****P < 0.0001. All statistical analysis was performed using R, GraphPad Prism, Python, or MATLAB.

## Reporting summary

Further information on research design is available in the Nature Portfolio Reporting Summary linked to this article.

## Data availability

Single-cell RNA-sequencing data has been deposited in the Gene Expression Omnibus under accession number GSE292904. Larger datasets have been deposited in the Figshare database under the following DOI: 10.6084/m9.figshare.28111892. Any additional data not found in source data that support the findings of this study will be made available with no restrictions upon request. Source data are provided with this paper.

## Code availability

Code used in this study for MATLAB based analysis uses standard functions and is available from upon request.

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

## Acknowledgements

We are grateful to C. L. Warriner for sharing a prior iteration of the wheel turning task; H. Rodrigues for assistance designing and constructing behavioral equipment; D. Ng and D. Peterka for discussion and feedback on the manuscript; I. Shieren, G. Martins, and M. Correia for technical assistance; L. Hammond, D. Peterka, and H. Ibarra Avila for assistance with various types of imaging and imaging data analysis; and T.M. Jessell for invaluable enthusiasm and support when this project was devised. Imaging was performed with support from the Zuckerman Institute's Cellular Imaging platform, and the National Institute of Health (NIH 1S10OD023587). L.M.C. and A.N. were Hellen Hay Whitney Foundation Fellows. L.M.C. is currently supported by an NIH Pathway to Independence Award (1K99NS127857). A.N. was supported also supported by an NIH Pathway to Independence Award (K99NS118053). R.M.C. was funded by grants from the NIH (5U19NS104649) and the Simons-Emory International Consortium on Motor Control.

## Author contributions

L.M.C., A.N., R.M.C. and V.M. designed experiments and interpreted data; L.M.C. and A.N. performed experiments and analyzed data; L.T. collected and analyzed anatomical data; A.K. assisted in experimental optimization; R.S. assisted with tissue clearing; and M.D.K. performed all flow cytometry; and L.M.C. and R.M.C. wrote the manuscript.

## Competing interests

The authors declare no competing interests.
