## [Transparent Peer Review file · Nature Communications]

Corticothalamic Neurons in Motor Cortex Have a Permissive Role in Motor Execution

Corresponding Author: Dr Rui Costa

Version 0:

Reviewer comments:

Reviewer #1

(Remarks to the Author)

In "Corticothalamic neurons in motor cortex have a permissive role in motor execution", Carmona et al use a fast, calcium dependent activity screen and single-cell RNA sequencing (sc-RNA-seq) to identify motor cortical cell types whose activity increases with learning of a wheel-pulling task. From the diverse cell types so identified, they focused on corticothalamic (CT) neurons, showing that their activity is anticorrelated with wheel velocity and that stimulating them during task execution impairs success. The paper is technically and conceptually innovative in its approach and the data as presented tells a compelling story, but multiple technical and analytical issues, along with discrepancies with prior literature, complicate interpretation of the results.

Major concerns:

Firstly, I have concerns about the specificity of the Foxp2-cre line used in many of the experiments, which was motivated by its high level of expression in the late learning enriched e9 cluster in the CaMPARI/sc-RNA-seq experiments. The authors note that Foxp2 is also expressed in corticospinal neurons (CSNs). It is also, according Figure S2, highly expressed in Grp88+ neurons, which given their co-expression of Gad1, Gad2, and Slc32a1 appear to be inhibitory cells. The proportion of non-CT Foxp2+ cells labelled by their injections for imaging/optogenetics should be quantified. In their imaging experiments, the authors partially control for CSN Foxp2 expression by retrograde tdTomato labelling, but as no injection can guarantee complete coverage, some tdTomato-negative, GCaMP7f-positive CSNs may remain in their dataset. This is compounded by their shallow choice of minimum imaging depth (300 μ m), which will bias recordings towards the more superficially-located CSNs. Fortunately they confirm their imaging and activity tagging results with another CT line (Ntsr1-cre), but not the optogenetic experiments. These should also be repeated in Ntsr1-cre mice, or at the very least the implications of non-specific stimulation should be carefully discussed.

For the imaging and optogenetic experiments, the authors analyzed wheel velocity, but this is an indirect read-out of movement as multiple strategies can be used to achieve the same velocity (e.g. one large pull versus many small ones in rapid succession). This is particularly important for interpreting the optogenetic results. What do the mice actually do when the light is on? Do they cease movement, attempt to turn the wheel but feebly, or make aberrant/uncoordinated movements? Video would be informative, kinematic tracking even more so. As it is, it is impossible to distinguish the permissive interpretation favored by the authors from the possibility that supernaturally activating the motor system (including possibly CSNs, see paragraph above) induces aberrant movements that interfere with the task, especially since Figure S8 implies that mice briefly turn the wheel faster when the light comes on.

In slice experiments, the authors show disynaptic corticocortical inhibition of CSNs by CTs. This conflicts with the paper the authors cite as corroborating these findings (Yamawaki & Shepherd (2015), J Neurosci), which showed weak or no such inhibition. This may be due to the very long (100 ms) photostimulation pulses used, which are likely to engage highly polysynaptic local circuits, e.g. CTs could excite L5A (Kim et al (2014), J Neurosci), which could in turn inhibit CSNs. If such a disynaptic CT->interneuron->CSN circuit emerges with learning that would be a very interesting result, but this should be confirmed with more conventional slice photostimulation parameters (e.g. 1-5 ms pulses). There is also the possibility that, in vivo, CTs could inhibit CSNs transthalamically, e.g. via activation of the reticular nucleus inhibiting CSN-projecting thalamocortical neurons, which the experiments in the paper cannot rule out.

The reporting of statistical analyses lacks detail. There are no n numbers, measures of central tendency and dispersion, or test statistics reported in the text, only p values. The number of animals is reported for none of the experiments bar one (on line 164), and even then it is a range rather than a precise number. Furthermore, in many cases the authors used regular ANOVA where repeated-measures ANOVA would be more appropriate, i.e. any comparison of multiple measurements from the same animal (e.g. in Figs 3, 4, S8-10 light on or off should be a within-subjects factor and control vs hChR2 between-subjects).

Lastly, depolarization in the dendrites does not inevitably drive somatic activation (Ledergerber & Larkum (2010), J Neurosci) and subthreshold postsynaptic potentials are sufficient to drive CaMPARI photoconversion (Zolnik et al (2016), J Physiol). Hence both activity read-outs used may reflect changes in input to CTs as much as output from CTs. This is briefly mentioned in the limitations section, and then only for the calcium imaging, but ought to be discussed more carefully.

Minor comments:

Line 159 – that the light stimulus is random with respect to the behavior should be noted here

Line 217 – how was the significance of cross-correlations assessed? It's very easy to get significant-looking but spurious cross-correlations, so some sort of shuffling or pseudotrial analysis should be performed.

Lines 283-285 – the first test (percent success for control vs hChR2) is embedded in the two-way analysis (light on/off × control/hChR2), so should be reported as a main effect of the latter rather than a separate test

Line 332 – the decrease in velocity in Figure S10K-L is very subtle and might be statistically but not meaningfully significant. How did stimulation before movement initiation affect trial success?

Line 338 – How much does locomotion on a flat, unobstructed surface depend on motor cortex? A better control would be a naturalistic forelimb task with minimal training requirements that is known to be cortically dependent, e.g. the irregular ladder task (Metz & Whishaw (2002), J Neurosci Meth)

Line 351 – As with the wheel-pulling task, wheel velocity is a coarse read-out of locomotion and some quantification of kinematics should be performed to rule out subtle effects

Line 381 – Figure S13F shows example traces from a single cell. I would like to see more detailed quantification of intrinsic properties, i.e. summary statistics for all cells in the dataset

Line 393 – I see no difference in latencies with training in Figure 5J. The authors should clarify how they quantified this change

Lines 599-608 – the authors appear to have mixed up mL/uL and uL/nL in several places

Lines 672-673 – flux density (ideally through the cranial window, or better yet estimates of irradiance at different cortical depths) should be reported, rather than total flux, as this is important for comparison with other studies using CaMPARI

Line 727 – how were p values corrected in this analysis?

Line 767 – this is not a receiver operating characteristic (ROC) analysis, it's just thresholding

Reviewer #2

(Remarks to the Author)

Carmona and colleagues present a manuscript assessing the role of corticothalamic neurons from the primary motor cortex (M1 CT) in a specific motor task (pulling a wheel). Combining activity-dependent labeling and single-cell RNAseq, they first performed an unbiased screen of M1 cell types to identify their engagement in the task at early and late training stages and selected M1 CT neurons for further study. Surprisingly, they found that M1 CT activity decreases during late training stages, specifically during movement. Optogenetically activating M1 CT neurons during the motor task perturbed its execution, but had no effect during basic locomotion, suggesting that this decrease is permissive for learning. Finally, the authors dissected M1 circuit activity during the task and showed that M1 CT neurons drive feedforward inhibition of M1 corticospinal neurons, suggesting that the decrease in M1 CT activity during the motor task leads to a disinhibition of corticospinal neurons.

This study is based on an elegant set of experiments using state-of-the-art methodology. The authors take advantage of the spatio-temporal specificity of cell-specific mouse lines, local and retrograde virus injections, and optogenetics, to tease out the contribution of M1 CT neurons in their motor task. Their data convincingly support that the decrease in M1 CT activity is important for the execution of their motor task, and their RNAseq data also opens the door to further studies of the roles of other M1 cell populations in movement. However, there are still possible confounding factors arising from the experimental paradigm, leading to a possible over-interpretation of the data regarding the role of M1 CT neurons in actual learning. As such, the manuscript requires more analysis (or more experiments) to conclusively support its main message. It would also benefit from additional information on some aspects of the experimental paradigm and of the chosen analytical methods.

Major concerns

1) The authors show that optogenetic activation of M1 CT neurons during trial initiation leads to a decrease in pull velocity and distance, especially during late training, but already with a trend during early training (Fig. 3). However, a similar decrease is observed both in successful and unsuccessful trials (Fig. S8-9). In the case of early training, the global effect seen when all trials are merged seems even driven exclusively by unsuccessful trials, since there is no significant change (not even a trend) for successful trials (Fig. S9). The same observation can be drawn from optogenetically activating M1 CT neurons during trials: the effect is the same regardless of the success of the trial (Fig. 4 and S10). This begs the question whether M1 CT neurons are truly involved in motor learning, since there is no apparent difference between successful or unsuccessful trials.

This concern becomes stronger considering the definition of unsuccessful trials, and whether or not they include trials where the animals do not move the wheel at all. Indeed, the methods section mentions that a trial is initiated when an IR beam is crossed and that it is considered unsuccessful if it doesn't pass a velocity threshold, but doesn't state whether a minimum distance or velocity must be reached. Considering that animals sometimes randomly extend and retract their paws, this suggests that unsuccessful trials could also include unrelated movements, as well as "trials" where the animal just rested their paws on the wheel and didn't try pulling the wheel (instead of pulling very weakly). The fact that the average distance in unsuccessful trials barely passes 0.5 (cm?) also supports that possibility. If that is the case, then the involvement of M1 CT neurons into motor learning becomes even more questionable, since their activity changes in the same manner regardless of the actual movement, even possibly in the absence of movement.

The most parsimonious explanation for the lack of difference between successful and unsuccessful trials, though, would be that the decrease in M1 CT activity is related to the internal state of the animals, such as their level of arousal or attention. The protocol doesn't mention if the animals were water deprived, but that would be logical given the distribution of water at the wheel. If the animals were videorecorded during behavior with sufficiently high resolution, the easiest way to control for arousal would be to look at the size of pupil dilation across all sessions and compare successful and unsuccessful trials. If indeed there is a change in arousal, it could also potentially explain the difference between the wheel pulling and locomotion, as the motivation/attention to acquire water is not necessary in the latter (more discussion in (2)).

2) The authors have also tried to show that M1 CT neuron activity is critical for a specific trained motor skill by comparing locomotion as an alternative movement. However, locomotion is a very different type of behavior, especially if it is unconstrained, so these results alone are not sufficient to support that M1 CT neurons are involved in learning. The fact that they do not see a similar decrease in velocity does suggest that the changes observed during wheel pulling are not ubiquitous for every movement type. That said, unlike the motor task that targets only one forelimb, walking involves movements of all four limbs, which reduces the functional impacts of activating M1 CT neurons in one hemisphere only, so that the lack of behavioral effect does not conclusively exclude a role of M1 CT neurons in that type of movement. The finding that a portion of M1 CT neurons are suppressed during walking further hints that M1 CT neurons may still be involved in locomotion, and thus in different (possibly all) types of movement.

But even if the involvement of M1 CT neurons were restricted to the wheel task, this would not necessarily indicate that they are permissive for learning. Indeed, they could be instead involved in goal-directed motion and/or fine motor control, regardless of the learning status. To show that they are involved specifically in learning, one possibility would be to assess the activity of M1 CT neurons in animals that have mastered the wheel task and for whom there is no learning involved anymore, so when the percentage of successful trials and the maximum velocity have reached a plateau. If the same decrease in activity is observed upon movement as during the late training phase and the same behavioral effect is observed upon optogenetic activation, then it is likely that the neurons are important for the movements underlying the task, rather than the learning of the motor skill.

Taking (1) and (2) together, while the authors are very careful to describe the role of M1 CT neurons as "permissive" for learned motor execution, the authors should provide some additional arguments (data or otherwise) to support a role of the decrease in M1 CT neuronal activity specifically in learning.

3) The authors control for the differential effect of light across tissue depth by comparing each cluster to its own control. While some clusters are expected to be located within a small depth range (e.g. L2/3 IT), this is not the case for all of them (e.g. PV and SST interneurons are found across layers II-VI), which could lead to an underestimation of the changes in these populations during training. Did the authors control for that particular aspect?

Additionally, were the control animals run through the entire length of the experiment, and were they time matched to the early and late training animals?

4) While FoxP2-expressing neurons project predominantly to the thalamus, it is not all of them. The authors cleverly address that drawback for their imaging experiments by co-labeling FoxP2 neurons projecting to the spinal cord to restrict their calcium imaging to CT neurons. Did they use a similar approach for their optogenetic experiments, and did they confirm that the majority of the ChrR2-expressing FoxP2 neurons projected to the thalamus? This could otherwise be a confounding factor to assess the role of CT FoxP2 neurons.

In a similar vein and since the authors did an extensive mapping of the projections of FoxP2 neurons, it would be good to see a quantification of the non-thalamic projections in Fig. S3. Knowing the proportion of these neurons would help assess how large of a confounding factor they could be.

The topographic organization of the FoxP2 axons across the thalamus is a particularly interesting piece of data, and following up on which thalamic nuclei are involved would be highly interesting, as the authors suggest. Based on the

positioning of their cranial window and other landmarks, could the authors estimate to which thalamic region the neurons that they imaged or inhibited project? This could provide a clue regarding which thalamic nuclei are involved and the interaction between corticothalamic and thalamocortical projections in the circuit.

5) The finding of increased M1 CT neuronal activity during the late-stage non-movement period is quite striking (Figs. 1 and S6), which could contribute to the larger degree of suppression at the late stage of training and may be more directly associated with learning. It would be nice if authors could discuss how increased baseline M1 CT neuronal activities and increased inhibitory inputs onto corticospinal neurons contribute to regulating the learned motor skills (controlling the optimal timing of the action initiation, restricting unwanted/nonspecific movement, etc.) We believe that including this discussion would provide a more comprehensive perspective on the role of M1 CT neurons in this motor learning paradigm.

Minor concerns

All relevant figures: please add the velocity and distance unit in the figure and figure legend where appropriate.

Fig. S1: Given that UV has limited spread *in vivo*, could authors provide an accompanying quantitation of the estimated conversion rates across the layers? This information will benefit other researchers who want to adapt the authors' experimental paradigm to their studies.

Fig. S2A-B: The clusters are presented in a different order between Fig. S2A and S2B, which itself differs from the order in Fig. 1F. Unless there is a particular reason for that different order, we would recommend stating the cluster identity (i1, i2, etc) and rearranging the clusters following the order given in Fig. 1F, to make the comparison between Fig. 1F, S2A and S2B easier. Please also do so for the additional statistics presented in Table 2.

Fig. S2B: Presumably the M1 cell atlas is displayed on the X axis, but it should be stated explicitly. It is also unclear what the axes refer to (especially as the color key does not cover the full range of colors displayed in the table), so a more detailed description of that panel in the figure legend would be helpful.

Figs. 3D and E show a drastic decrease in the maximal velocity for the hChR2 group in both the traces and the paired line plot. However, in Figs S8B and E, when separated by whether the trials were successful, there was no evident reduction in maximal velocity in the traces of both cases. Furthermore, in all cases, the values of paired line plots were higher than those in the traces. Could the authors address what might cause these discrepancies or at least clarify how the velocity measure (and other behavioral measures used in all figures) was quantified in the method section?

Fig. S9C/F: Three animals show a negative distance, including in a successful trial. Did the animals push the wheel instead of pulling it? Shouldn't they be excluded in that case, since that movement wasn't part of training?

Line 171: Please include the references to the published annotations.

Lines 183-185: please specify in the text which cluster numbers the authors referred to (see line 189) for readers to better cross-reference Figure 1.

Line 266: "Decrease" should read "decreased" or "decrease in".

Line 595: "Later" should read "lateral".

Lines 599 – 610: The units for the virus volumes are inconsistent. mL and, in some places, uL seem to be used by error throughout the section. Please double-check and correct the typos accordingly.

Reviewer #3

(Remarks to the Author)

Version 1:

Reviewer comments:

Reviewer #1

(Remarks to the Author)

The revisions satisfactorily address most of the issues. The quantification of FoxP2 spillover into corticospinals, replication of the behavioral experiments in Ntsr1-cre mice, quantification of kinematics, and analysis of excitation and inhibition latencies strengthen the conclusions of this thorough and intriguing study. The newly included supplementary movies are informative. There are a few remaining concerns, primarily about clarification of the new experiments and analyses.

It is not clear how exactly were the data shuffled for the shuffling analysis. Were fluorescence traces completely scrambled, or were they shuffled in such a way as to preserve the autocorrelation structure while breaking the temporal relation to wheel velocity, e.g. trial shuffling or session shuffling? How many shuffling iterations were performed?

The inclusion of descriptive statistics for each group in each ANOVA is excessive and hampers readability. Move these to

the tables showing the p-values of pairwise comparisons, and report descriptive statistics in the text only for two-group tests (e.g. t-tests). These tables should also include standard ANOVA summary tables (degrees of freedom, sums of squares, etc. for each effect tested and residual error, plus F statistics and p-values for effects), and should also specify which effects were modelled as within or between subjects. Finally, almost all numbers in the text are reported to an excessive number of significant figures. Two to three is enough in most cases.

Regarding my previous point about stimulation before movement initiation, reporting percent success for category 1 and category 2 trials from Rebuttal Figure 6 would address this concern, even if the number of trials is low. It can still be calculated if there are more than zero trials, which is the case for most mice. It would be fine to pool these two categories to help increase the trial numbers.

Minor comments:

More raw data should be shown for the quantification of kinematics, e.g. average traces similar to those for wheel velocity, before jumping to summary statistics.

Where percentage of successful trials that had light on is reported, the text changes are helpful, but in the figures (Figs 3B,H; S9B; S11B), a change of y-axis label from “% of success” to e.g. “% light on (successful trials)” would make things even clearer.

Line 305: should read “decrease of the maximum distance traveled by the right forelimb”.

Line 434: should read “stimulation of M1CT evoked relatively small excitatory postsynaptic currents”

Fig S3: the A/P profiles of retrograde cortical labeling from thalamus in Fig S3F do not seem to match the 2D profiles in Fig S3G. Are the medial and lateral labels flipped in Fig S3G?

Fig S4: the quantification in Fig S4C is useful, but looking at the example in Fig S4B, there appear to be one or two GFP+, tdTomato -ve cells in layer 5 (based on having similar depth to tdTomato +ve corticospinal cells). If the layer boundaries can be accurately determined from the bright field images of these slices, an even better quantification would be L6 GFP+ vs L5 GFP+/tdTom+ vs L5 GFP+/tdTom-, to estimate the proportion of non-retrogradely-labelled FoxP2 +ve corticospinals that could be contaminating the fluorescence data.

Fig S14: panels K and L should use non parametric statistics (i.e. median and median absolute deviation or IQR) due to the highly skew distributions evident in panels I and J. Calculating EPSC to IPSC latency differences per-cell before averaging would also help by accounting for cell-to-cell variability.

Reviewer #2

(Remarks to the Author)

The authors have fully addressed our concerns and we support the publication of this excellent study.

Reviewer #3

(Remarks to the Author)

See below.

We thank the reviewers for finding our findings interesting and novel and for the constructive and insightful feedback on our manuscript. We have addressed all concerns with new experiments, new analyses, or changes in the text. This has resulted in 1 entirely new figure (Supplementary Figure 9), 14 new figure panels, and 4 Supplementary Movies. We have also made numerous changes throughout the text and methods (highlighted in yellow) and have provided more detailed descriptions of statistics where appropriate. We believe these changes have added clarity to our manuscript and strengthen the arguments presented throughout.

We hope that these changes now make the manuscript suitable for publication in Nature Communications. Please find below a point-by-point answer to each of the reviewer's comments (reviewer comment in black and our response in blue).

Point by point address of reviewers' points

Reviewer #1 (Remarks to the Author):

In "Corticothalamic neurons in motor cortex have a permissive role in motor execution", Carmona et al use a fast, calcium dependent activity screen and single-cell RNA sequencing (sc-RNA-seq) to identify motor cortical cell types whose activity increases with learning of a wheel-pulling task. From the diverse cell types so identified, they focused on corticothalamic (CT) neurons, showing that their activity is anticorrelated with wheel velocity and that stimulating them during task execution impairs success. The paper is technically and conceptually innovative in its approach and the data as presented tells a compelling story, but multiple technical and analytical issues, along with discrepancies with prior literature, complicate interpretation of the results.

Major concerns:

Firstly, I have concerns about the specificity of the Foxp2-cre line used in many of the experiments, which was motivated by its high level of expression in the late learning enriched e9 cluster in the CaMPARI/sc-RNA-seq experiments. The authors note that Foxp2 is also expressed in corticospinal neurons (CSNs). It is also, according Figure S2, highly expressed in Grp88+ neurons, which given their co-expression of Gad1, Gad2, and Slc32a1 appear to be inhibitory cells. The proportion of non-CT Foxp2+ cells labelled by their injections for imaging/optogenetics should be quantified. In their imaging experiments, the authors partially control for CSN Foxp2 expression by retrograde tdTomato labelling, but as no injection can guarantee complete coverage, some tdTomato-negative, GCaMP7f-positive CSNs may remain in their dataset. This is compounded by their shallow choice of minimum imaging depth (300 um), which will bias recordings towards the more superficially-located CSNs. Fortunately they confirm their imaging and activity tagging results with another CT line (Ntsr1-cre), but not the optogenetic experiments. These should also be repeated in Ntsr1-cre mice, or at the very least the implications of non-specific stimulation should be carefully discussed.

Thank you for the opportunity to clarify this point. First, it is important to note that these FoxP2+ neurons in layer V are corticospinal neurons but also project to thalamus with similar patterns to those of layer VI neurons (Figure S3). Thus, the FoxP2 population is likely predominantly corticothalamic: another way to view this would be that some corticothalamic neurons have cell bodies in layer 5 and also project to the spinal cord.

Second, we quantified the number of layer V FoxP2+ neurons by counting the number of corticospinal neurons labeled with the retrograde spinal injection in our calcium imaging mice relative to the number of layer VI corticothalamic neurons, expressing only GCaMP7f. In these animals, layer V corticospinal neurons make up about 11% of FoxP2+ neurons labeled (Rebuttal Figure 1, manuscript S4C).

Rebuttal Figure 1. Percent of all FoxP2 labeled neurons (GCaMP7f expression if FoxP2-cre mice) that are single positive (GCaMP7f, L6) or double positive (GCaMP7f and retrograde, cre-dependent TdTomato injected into cervical spinal cord); L6: 89.2 ± 5.65 , L5: 10.82 ± 5.65 , N=3.

Third, as the reviewer suggested, to further address this concern in our optogenetic experiments, we have repeated the manipulation at late learning with light delivered concurrent with trial start in Ntsr1-cre animals. As with our FoxP2 animals, we observed a decrease in the ability of opsin animals to execute the task during light on trials (Rebuttal Figure 2B, C). This is accompanied by a decrease in velocity light on trials in opsin animals but not control animals (Rebuttal Figure 2D, E). These data has also been added to the manuscript (Figure S9).

Rebuttal Figure 2. **A** Schematic of optogenetic light delivery for closed loop at trial start at late training for wheel turning task in Ntsr1-cre animals. Light was pulsed at 20 hz; 10 ms pulse width. **B** Percent of successful trials with light on (# successful trials during light on / # of all successful trials); control, gray; hChR2, blue; error bars, SEM; two-tailed unpaired t-test, $p=0.0357$. For B-F: N=4, control group; N=6, hChR2 group. Each point is average of all light delivery sessions. **C** Percentage of trials that are successful for each animal with light on or off in each group (# successful trials during light on(off) / # of all trials during light on(off)); error bars, SEM; two-way ANOVA, multiple comparisons; $p=0.5302$, control; $p=0.0491$, hChR2, $p=0.3886$ for group x condition. **D** Velocity traces of all trials from control, top, and hChR2, bottom, animals during light, blue, and no light trials, gray; 0=trial start; blue bar denotes time of light, $p>0.9999$ (control), $p<0.0001$ (hChR2), time x condition, two way ANOVA; solid line, mean; shaded area, SEM. **E** Maximum velocity of all trials for each animal with light on or off in each group; two-way ANOVA, multiple comparisons, $p=0.5488$ (control), $p=0.0452$ (hChR2), $p=0.3603$ for group x condition; **F** Wheel distance traveled during all trials for each animal with light on or off for each group; two way ANOVA, multiple comparisons, $p=0.8619$, control, $p=0.0997$, hChR2, $p=0.3203$ for group x condition.

For the imaging and optogenetic experiments, the authors analyzed wheel velocity, but this is an indirect read-out of movement as multiple strategies can be used to achieve the same velocity (e.g. one large pull versus many small ones in rapid succession). This is particularly important for interpreting the optogenetic results. What do the mice actually do when the light is on? Do they cease movement, attempt to turn the wheel but feebly, or make aberrant/uncoordinated movements? Video would be informative, kinematic tracking even more so. As it is, it is impossible to distinguish the permissive interpretation favored by the authors from the possibility that supernaturally activating the motor system (including possibly CSNs, see paragraph above) induces aberrant movements that interfere with the task, especially since Figure S8 implies that mice briefly turn the wheel faster when the light comes on.

We have attempted to track the right forelimb using DeepLabCut¹. We have exacted the maximum distance of forelimb movement during light on and light off trials. As in our analysis of the velocity of the wheel encoder, we observe a significant decrease in distance traveled by the forelimb in hChR2 animals in light on trials (Rebuttal Figure 3 and manuscript Figure S8H). No difference is observed in control animals. This supports a cease in movement rather than aberrant movements. We have also included some sample videos that demonstrate this phenotype (Supplementary Movie 1-4). We believe that the high velocity observed in successful, light on trials of hChR2 animals reflects a selection for a particular execution strategy that allows mice to rapidly pull the wheel before the full effect of the optogenetic stimulation sets in. This is supported by a change in the proportion of successful to unsuccessful light on trials in opsin mice (Rebuttal Table 1). Unfortunately, our videos are taken at 150 frames per second, and are from a single angle precluding more extensive analysis of limb trajectories or more optimal 3D analysis (in many trials mice execute the pulls within 200 ms).

Rebuttal Figure 3. Max distance traveled by forelimb from start of trial. Position extracted using DeepLabCut and the Euclidian distance was calculated to each subsequent frame within the trial. The max was taken for each trial and averaged for each animal for either light on or light off trials; two-way ANOVA, multiple comparisons; p=0.1951, control; p=0.0088, hChR2; p=0.1860 for group x condition; N=7, control; N=6, hChR2;

	on		off	
	unsuccessful	successful	unsuccessful	successful
control	0.742	0.258	0.711	0.289
opsin	0.8782	0.1218	0.677	0.323

Rebuttal Table 1. Proportion of unsuccessful and successful trials executed with light on or off for control and opsin groups.

References:

1 Mathis, A. *et al.* DeepLabCut: markerless pose estimation of user-defined body parts with deep learning. *Nat Neurosci* **21**, 1281-1289 (2018). <https://doi.org/10.1038/s41593-018-0209-y>

In slice experiments, the authors show disynaptic corticocortical inhibition of CSNs by CTs. This conflicts with the paper the authors cite as corroborating these findings (Yamawaki & Shepherd (2015), *J Neurosci*), which showed weak or no such inhibition. This may be due to the very long (100 ms) photostimulation pulses used, which are likely to engage highly polysynaptic local circuits, e.g. CTs could excite L5A (Kim et al (2014), *J Neurosci*), which could in turn inhibit CSNs. If such a disynaptic CT->interneuron->CSN circuit emerges with learning that would be a very interesting result, but this should be confirmed with more conventional slice photostimulation parameters (e.g. 1-5 ms pulses). There is also the possibility that, in vivo, CTs could inhibit

CSNs transthalamically, e.g. via activation of the reticular nucleus inhibiting CSN-projecting thalamocortical neurons, which the experiments in the paper cannot rule out.

The reviewer correctly points out that Yamawaki & Shepherd observe small inhibitory postsynaptic currents evoked in PT neurons after photostimulating CT neurons that express ChR2 (around 15-fold less, although the exact mean evoked amplitudes were not reported), compared to other local microcircuit connectivity. Our results in untrained mice fully agree with this result, despite the longer photostimulation period we used.

We also acknowledge that *any* photostimulation protocol alone (even < 5ms) cannot isolate monosynaptic connectivity; to do so requires pharmacological manipulations (i.e., TTX, high divalent ACSF, etc.) that would inevitably block feedforward inhibition. Therefore, we remain agnostic to the exact nature of the feedforward inhibitory microcircuit (two synapses versus three synapses, etc.) that is potentiated following learning.

Still, we agree with the reviewer that more analysis of the latency of evoked responses is interesting and contributes to the merit of our study, so we have performed the following analysis.

For each CSN targeted, we identified the first timepoint at which average ChR2-evoked EPSCs or IPSCs exceeded a conservative threshold of 3 times the standard deviation of a baseline period (defined as 8ms surrounding the onset of the light stimulus). (Rebuttal Figure 4). Using this method, we detect significant responses within 7 ms and 17 ms of stimulus onset for excitatory and inhibitory currents, respectively (Rebuttal Figure 4). In the post-training neurons, we observe similar latencies with significant excitatory responses within 6 ms of stimulus onset and inhibitory currents within 13 ms of stimulus onset (Rebuttal Figure 4). This result indicates that the potentiation of feedforward inhibition following learning operates at a similar timescale to the (comparatively small) feedforward inhibition seen in naïve mice. As an alternative analysis and to directly compare the onset of inhibitory currents in neurons from untrained and trained mice, we binned the responses into 0.5 ms windows and performed unpaired t-test across these windows. We determined that the timepoint they first significantly diverge in at 12 ms (Rebuttal Figure 4). Thus, the augmented feedforward inhibition seen following learning is not due to the recruitment of polysynaptic connectivity that could have emerged several tens of milliseconds following stimulus onset. Instead, this timescale is consistent with a short latency feedforward microcircuit. We now include these latency analyses in our study as supplemental results (Figure S14G-M). Finally, we have also noted the potential for a transthalamic regulatory circuit in the discussion (lines 522-524).

Rebuttal Figure 4. **A-D** Responses of corticospinal neurons after stimulation of corticothalamic neurons; smoothed 2ms sliding window. Excitatory, black; inhibitory, gray. **C, D** zoom in of **A, B**, respectively; dots indicate time when response is at least 3 SD above the baseline period; green excitatory; red, inhibitory. **A,C** untrained, **B,D** trained. **E,F** Quantification of latency of excitatory and inhibitory responses post stimulus. **E**, untrained; **F**, trained; dot, mean; range, SEM. **G** Comparison of inhibitory responses in neurons from untrained, lower gray line, and trained, upper green line, binned over 0.5 ms. Each bar denotes p-value of each set of values at that timepoint. Light green bar, first significant timepoint.

The reporting of statistical analyses lacks detail. There are no n numbers, measures of central tendency and dispersion, or test statistics reported in the text, only p values. The number of animals is reported for none of the experiments bar one (on line 164), and even then it is a range rather than a precise number. Furthermore, in many cases the authors used regular ANOVA where repeated-measures ANOVA would be more appropriate, i.e. any comparison of multiple measurements from the same animal (e.g. in Figs 3, 4, S8-10 light on or off should be a within-subjects factor and control vs hChR2 between-subjects).

We have added mean and standard deviation values for all groups where p-values are reported within the text. n and N values had been reported in figure legends but have now also been added to the text. We also apologize for the lack of clarity but where appropriate, the ANOVA reported had been for repeated measures with corrections for multiple comparisons. We have noted this where appropriate, and we have also added the group x condition p values.

Lastly, depolarization in the dendrites does not inevitably drive somatic activation (Ledergerber & Larkum (2010), J Neurosci) and subthreshold postsynaptic potentials are sufficient to drive CaMPARI photoconversion (Zolnik et al (2016), J Physiol). Hence both activity read-outs used may reflect changes in input to CTs as

much as output from CTs. This is briefly mentioned in the limitations section, and then only for the calcium imaging, but ought to be discussed more carefully.

We have added this important point to our discussion of the limitations of our CaMPARI based approach (lines 561-564).

Minor comments:

Line 159 – that the light stimulus is random with respect to the behavior should be noted here

A sentence has been added (lines 160-161) to address this.

Line 217 – how was the significance of cross-correlations assessed? It's very easy to get significant-looking but spurious cross-correlations, so some sort of shuffling or pseudotrial analysis should be performed.

Thank you. We have added shuffling for this analysis. By shuffling the neuronal signal and assessing over the session or trials, as in the previously reported cross-correlation analyses, we do not observe a relationship between the neuronal signal and the wheel velocity. We have added the panels in Figure S4D, S4G, S12B, and S12F of the revised manuscript. The direct comparisons can also be seen below in Rebuttal Figure 5.

Rebuttal Figure 5. **A** Cross-correlation of Z-scored calcium $\Delta F/F$ for each unit of corticothalamic neurons in FoxP2 animals to wheel velocity for whole session; mean, solid gray line; shaded area, SEM; **B** Cross-correlation of shuffled Z-scored calcium $\Delta F/F$ for each unit of corticothalamic neurons in FoxP2 animals to wheel velocity for whole session; mean, solid gray line; shaded area. **C** Cross-correlation of Z-scored calcium $\Delta F/F$ to wheel velocity for whole session of each group of corticothalamic neurons from FoxP2-cre animals during wheel pulls; solid lines, mean; shaded areas, SEM; wheel velocity during pulls. **D** Cross-correlation for same groups as in C but with shuffled Z-scored calcium $\Delta F/F$. **E** Cross-correlation of Z-scored calcium $\Delta F/F$ for each unit of corticospinal neurons to wheel velocity for whole session; mean, solid gray line; shaded area,

SEM. **F** Cross-correlation of shuffled Z-scored calcium $\Delta F/F$ for each unit of corticospinal neurons to wheel velocity for whole session; mean, solid gray line; shaded area. **F** Cross-correlation of Z-scored calcium $\Delta F/F$ for corticospinal neurons to wheel velocity for pulls at early (blue) and late (green) training sessions; solid line, mean; shaded area, SEM. **F** Cross-correlation of shuffled Z-scored calcium $\Delta F/F$ for corticospinal neurons to wheel velocity for wheel pulls; mean, solid gray line; shaded area.

Lines 283-285 – the first test (percent success for control vs hChR2) is embedded in the two-way analysis (light on/off \times control/hChR2), so should be reported as a main effect of the latter rather than a separate test

We apologize for the lack of clarity in these plots and our incorrect description which has been edited throughout. Percent of successful trials with light on (as in Figure 3B) compares: # successful trials during light on / # of all successful trials, whereas percent of trials that are successful for each animal with light on or off in each group (as in Figure 3C) compares: # successful trials during light on (OR off) / # of all trials with light on (OR off). These are separate comparisons so we have left the plots as before, but we have added the corresponding equation to the text and legends.

Line 332 – the decrease in velocity in Figure S10K-L is very subtle and might be statistically but not meaningfully significant. How did stimulation before movement initiation affect trial success?

To examine trial initiation, we delivered light randomly through the session. Although the number of trials was comparable to other cohorts, the probability of capturing light on trials was low. We, thus, cannot compare trial success. Instead, we examined wheel pulls similar to those that would have been executed during trials (as noted in lines 366-367). These were post hoc filtered and aggregated (S11K-L). In addition, we have examined the total number of trials executed during light on or off periods throughout the session (Rebuttal Figure 6). Here the session is broken up into 1.2 second time intervals (the duration of light delivery). Intervals with light on are then compared to a subset of the intervals with light off (subsampling to yield comparable trial numbers as light was only delivered for 1/10th of the session). We observe no difference in the number of trials detected in these light on or off intervals.

Rebuttal Figure 6. Percent of all trials, as depicted in each category (S, green, trial start; R, red, trial end, blue bar, light on period), that fall within a period of light on or a subset of light off periods (subsampling to match total time) for control or hChR2 mice.

Line 338 – How much does locomotion on a flat, unobstructed surface depend on motor cortex? A better control would be a naturalistic forelimb task with minimal training requirements that is known to be cortically dependent, e.g. the irregular ladder task (Metz & Whishaw (2002), J Neurosci Meth)

We performed these experiments as a control precisely because we did not expect walking to be heavily M1 dependent, we thought this would allow us to rule out any indirect effects of stimulation of these neurons that would perturb movement in general. Importantly, we had observed that about half of M1^{CT} neurons in FoxP2-cre mice imaged in the caudal forelimb area decrease in activity during walking implicating a similar decrease in activity of these neurons during locomotion, and so these experiments are meant to show that these optogenetic stimulations do not affect every movement.

Line 351 – As with the wheel-pulling task, wheel velocity is a coarse read-out of locomotion and some quantification of kinematics should be performed to rule out subtle effects

We have attempted to track the right forelimb during locomotion trials using DeepLabCut¹. Because the mice are walking when trials begin and the movement is not aligned, we have calculated the total forelimb movement during the light on period or equivalent control trials. We have excluded any trials in which the probability of accurate tracking was less than 80% in more than 10% of frames for each trial period including 200 ms before and after the trial. Animals with less than 3 trials have been excluded. As with our examination of the encoder velocity, we do not see a difference in forelimb movement when comparing light on or light off trials in either the control or opsin group (Rebuttal Figure 7).

Rebuttal Figure 7. Total distance traveled by forelimb during walking trials. Position was extracted using DeepLabCut¹, and the Euclidian distance was from frame to frame and summed. All trials were averaged for each animal for either light on or light off trials; two-way ANOVA, multiple comparisons; p=0.9924, control; p=0.8482, hChR2, p=0.776 for group x condition; N=6 control; N=6, hChR2;

References:

1 Mathis, A. *et al.* DeepLabCut: markerless pose estimation of user-defined body parts with deep learning. *Nat Neurosci* **21**, 1281-1289 (2018). <https://doi.org/10.1038/s41593-018-0209-y>

Line 381 – Figure S13F shows example traces from a single cell. I would like to see more detailed quantification of intrinsic properties, i.e. summary statistics for all cells in the dataset

We focused this work on patching CS neurons while stimulating CT neurons. This included Cs and QX-314 in the pipette for voltage clamp, blocking Na channels for recording inhibitory currents and precluding us from characterizing intrinsic properties of patched CS neurons.

Line 393 – I see no difference in latencies with training in Figure 5J. The authors should clarify how they quantified this change

We apologize for the incorrect use of this term. We had wanted to highlight the difference in profiles of the inhibitory responses before and after training soon after stimulation. We have examined the latencies of the

responses (Rebuttal Figure 4), as the reviewer notes, they are similar for trained and untrained neurons. We have removed this statement from the manuscript. However, the magnitude of the inhibitory responses of trained neurons diverge from those of the untrained neurons within 12 ms of the stimulus onset (Rebuttal Figure 4).

Lines 599-608 – the authors appear to have mixed up mL/uL and uL/nL in several places

Thank you for the careful read of the methods! We have fixed these.

Lines 672-673 – flux density (ideally through the cranial window, or better yet estimates of irradiance at different cortical depths) should be reported, rather than total flux, as this is important for comparison with other studies using CaMPARI

We have estimated irradiance at the brain surface to 1mm of cortical depth (Rebuttal Table 2 and Rebuttal Figure 8). The parameters and equations used are also outlined below. We have modified the Methods section to include flux density at the cranial window as well as the table and the outline of our calculations.

cortical depth (um)	Irradiance (mW/mm²)
100 um	49.282
200 um	29.299
300 um	17.419
400 um	10.356
500 um	6.157
600 um	3.66
700 um	2.176
800 um	1.294
900 um	0.769
1000 um	0.457

Rebuttal Table 2 and Rebuttal Figure 8. Irradiance calculated at the cranial window and at every 100 um of cortical depth until 1mm. The experimental parameters and equations utilized are listed below.

Experimental parameters:

LED light source output: 60 mW

Light wavelength: 405 nM

Fiber diameter: 960 uM

Fiber NA: 0.63

Irradiance at the fiber tip (E_0):

$$E_0 = \frac{P}{\pi\left(\frac{d}{2}\right)^2}$$

P : Power = 60 mW = 0.06W

d : diameter of fiber = 960 uM = 0.00096 m

$$E_0 = \frac{0.06}{\pi\left(\frac{0.00096}{2}\right)^2}$$

$$E_0 = 82893 \text{ W}/m^2$$

Irradiance at various cortical depths:

$$E(z) = E_0 \cdot e^{-\mu_t z}$$

z : depth

μ_t : total attenuation coefficient

To calculate the total attenuation coefficient (μ_t):

$$\mu_t = \mu_a + \mu_s$$

μ_a : absorption coefficient
estimated to be 0.2 mm^{-1}

μ_s : scattering coefficient
estimated to be between 2 mm^{-1}

estimates from reference:

- 1 Yaroslavsky, A. N. *et al.* Optical properties of selected native and coagulated human brain tissues in vitro in the visible and near infrared spectral range. *Phys Med Biol* **47**, 2059-2073 (2002).
<https://doi.org/10.1088/0031-9155/47/12/305>

$$\begin{aligned}\mu_t &= 0.2 + 5 \\ \mu_t &= 5.2 \text{ mm}^{-1}\end{aligned}$$

Sample calculation for 100 um below the surface:

$$\begin{aligned}E(100\mu\text{m}) &= E_0 \cdot e^{-\mu_t z} \\ E(100\mu\text{m}) &= 82893 \cdot e^{-5200(0.0001)} \\ E(100\mu\text{m}) &= 82893 \cdot e^{-0.52} \\ E(100\mu\text{m}) &= 82893 \cdot 0.5945 \\ E(100\mu\text{m}) &= 49,281.59\end{aligned}$$

Line 727 – how were p values corrected in this analysis?

The p values are FDR (Benjamini-Hochberg) corrected. We have added this to the methods (line 809).

Line 767 – this is not a receiver operating characteristic (ROC) analysis, it's just thresholding

We have reassessed our analysis, and the reviewer is correct. We edited the manuscript to reflect this.

Reviewer #2 (Remarks to the Author):

Carmona and colleagues present a manuscript assessing the role of corticothalamic neurons from the primary motor cortex (M1 CT) in a specific motor task (pulling a wheel). Combining activity-dependent labeling and single-cell RNAseq, they first performed an unbiased screen of M1 cell types to identify their engagement in the task at early and late training stages and selected M1 CT neurons for further study. Surprisingly, they found that M1 CT activity decreases during late training stages, specifically during movement. Optogenetically activating M1 CT neurons during the motor task perturbed its execution, but had no effect during basic locomotion, suggesting that this decrease is permissive for learning. Finally, the authors dissected M1 circuit activity during the task and showed that M1 CT neurons drive feedforward inhibition of M1 corticospinal neurons, suggesting that the decrease in M1 CT activity during the motor task leads to a disinhibition of corticospinal neurons.

This study is based on an elegant set of experiments using state-of-the-art methodology. The authors take advantage of the spatio-temporal specificity of cell-specific mouse lines, local and retrograde virus injections, and optogenetics, to tease out the contribution of M1 CT neurons in their motor task. Their data convincingly support that the decrease in M1 CT activity is important for the execution of their motor task, and their RNAseq data also opens the door to further studies of the roles of other M1 cell populations in movement. However, there are still possible confounding factors arising from the experimental paradigm, leading to a possible over-interpretation of the data regarding the role of M1 CT neurons in actual learning. As such, the manuscript requires more analysis (or more experiments) to conclusively support its main message. It would also benefit from additional information on some aspects of the experimental paradigm and of the chosen analytical methods.

Major concerns

1) The authors show that optogenetic activation of M1 CT neurons during trial initiation leads to a decrease in pull velocity and distance, especially during late training, but already with a trend during early training (Fig. 3). However, a similar decrease is observed both in successful and unsuccessful trials (Fig. S8-9). In the case of early training, the global effect seen when all trials are merged seems even driven exclusively by unsuccessful trials, since there is no significant change (not even a trend) for successful trials (Fig. S9). The same observation can be drawn from optogenetically activating M1 CT neurons during trials: the effect is the same regardless of the success of the trial (Fig. 4 and S10). This begs the question whether M1 CT neurons are truly involved in motor learning, since there is no apparent difference between successful or unsuccessful trials.

We agree that our data does not provide evidence for the role of M1^{CT} neurons in motor learning, and apologize if the language we used led to a misunderstanding. Rather, like the reviewer, we think M1^{CT} neurons play a role in motor execution, particularly in forelimb tasks that are well trained/skilled. The optogenetic manipulations have a similar effect on both successful and unsuccessful trials: a decrease in wheel velocity. Because we see this similar effect regardless of the maximum velocity of a trial, we believe this indicates a deficit in motor execution. Given that a velocity threshold must be surpassed for success, this led to a decrease in the proportion of successful trials in the opsin group. In other words, we did not determine whether light would be given during a successful or unsuccessful trial. Instead, we randomly deliver light on a subset of trials and posthoc analyze whether it was successful or unsuccessful. Thus, a change in the proportion of successful to unsuccessful trials is indicative of a deficit in achieving the threshold velocity for reward. When manipulations are performed for ongoing trials, a change in success is not expected as there is sufficient time for mice to reach threshold velocity for reward. However, we again observe a decrease in velocity once the light is turned on again indicating a deficit in motor execution. We now clarify throughout that this manuscript identifies a permissive role of M1^{CT} neurons in motor execution!

This concern becomes stronger considering the definition of unsuccessful trials, and whether or not they include trials where the animals do not move the wheel at all. Indeed, the methods section mentions that a trial is initiated when an IR beam is crossed and that it is considered unsuccessful if it doesn't pass a velocity threshold, but doesn't state whether a minimum distance or velocity must be reached. Considering that animals sometimes randomly extend and retract their paws, this suggests that unsuccessful trials could also include unrelated movements, as well as "trials" where the animal just rested their paws on the wheel and didn't try

pulling the wheel (instead of pulling very weakly). The fact that the average distance in unsuccessful trials barely passes 0.5 (cm?) also supports that possibility. If that is the case, then the involvement of M1 CT neurons into motor learning becomes even more questionable, since their activity changes in the same manner regardless of the actual movement, even possibly in the absence of movement.

We apologize for this incomplete description of the task. We have edited the description in the methods section “Behavior” (lines 700-703). A trial would only commence if the IR beam break and capacitance sensor on the wheel rungs each passed a threshold. A velocity threshold was also set to exclude minor movements of the wheel and a higher velocity threshold determined the success of the trial.

The most parsimonious explanation for the lack of difference between successful and unsuccessful trials, though, would be that the decrease in M1 CT activity is related to the internal state of the animals, such as their level of arousal or attention. The protocol doesn’t mention if the animals were water deprived, but that would be logical given the distribution of water at the wheel. If the animals were videorecorded during behavior with sufficiently high resolution, the easiest way to control for arousal would be to look at the size of pupil dilation across all sessions and compare successful and unsuccessful trials. If indeed there is a change in arousal, it could also potentially explain the difference between the wheel pulling and locomotion, as the motivation/attention to acquire water is not necessary in the latter (more discussion in (2)).

We unfortunately do not have videos of sufficient resolution to examine pupil dilation. However, we have performed open loop manipulations (Figures S10J-M) that encompass trial start. If activation of these neurons were to have an effect on the arousal state of the animal, we would predict that they would perform less trials when the light was on prior to trial initiation. We, therefore, examined the proportion of these trials as a fraction of all trials in a session (Rebuttal Figure 9). Here, we do not observe a difference in the proportion of trials executed with light on or off conditions as broken down into any of the categories that include the light on prior to trial start (category 1, 2, and 4).

Rebuttal Figure 9. Percent of all trials, as depicted in each category (S, green, trial start; R, red, trial end, blue bar, light on period), that fall within a period of light on or a subset of light off periods (subsampling to match total time) for control or hChR2 mice. Note this figure is the same as Rebuttal Figure 6.

2) The authors have also tried to show that M1 CT neuron activity is critical for a specific trained motor skill by comparing locomotion as an alternative movement. However, locomotion is a very different type of behavior, especially if it is unconstrained, so these results alone are not sufficient to support that M1 CT neurons are involved in learning. The fact that they do not see a similar decrease in velocity does suggest that the changes observed during wheel pulling are not ubiquitous for every movement type. That said, unlike the motor task that targets only one forelimb, walking involves movements of all four limbs, which reduces the functional impacts of activating M1 CT neurons in one hemisphere only, so that the lack of behavioral effect does not

conclusively exclude a role of M1 CT neurons in that type of movement. The finding that a portion of M1 CT neurons are suppressed during walking further hints that M1 CT neurons may still be involved in locomotion, and thus in different (possibly all) types of movement.

Again, we want to repeat that our intent was never to claim M1^{CT} neurons are involved in learning. It is important to first note that our locomotion assay involved head fixation to allow us to deliver light through the cranial window. This is noted in the methods section “Behavior” (line 710).

We performed these experiments as a control precisely because we did not expect walking to be heavily M1 dependent, we thought this would allow us to rule out any indirect effects that could generally perturb movement (please see also response to reviewer 1). Furthermore, we have focused our examination on the caudal forelimb area of M1 which should encompass many of the neurons contributing to movement of the right forelimb. Because small deficits could be missed by compensation of the other limbs, we have attempted to examine the kinematics of the right forelimb during light on or off trials using DeepLabCut¹. As mice are walking when trials begin and the movement is not aligned, we have calculated the total forelimb movement during the light on period or equivalent control trials. We have excluded any trials in which the probability of accurate tracking was less than 80% in more than 10% of frames for each trial period including 200 ms before and after the trial. Animals with less than 3 trials have been excluded. As with our examination of the encoder velocity, we do not see a difference in forelimb movement when comparing light on or light off trials in either the control or opsin group (Rebuttal Figure 10) providing further evidence for the lack of a strong deficit during locomotion.

Rebuttal Figure 10. Total distance traveled by forelimb during walking trials. Position was extracted using DeepLabCut¹, and the Euclidian distance was from frame to frame and summed. All trials were averaged for each animal for either light on or light off trials; two-way ANOVA, multiple comparisons; $p=0.9924$, control; $p=0.8482$, hChR2 $N=6$ control; $p=0.776$ for group x condition; $N=6$, hChR2. Note this figure is the same as Rebuttal Figure 7.

References:

- 1 Mathis, A. *et al.* DeepLabCut: markerless pose estimation of user-defined body parts with deep learning. *Nat Neurosci* **21**, 1281-1289 (2018). <https://doi.org/10.1038/s41593-018-0209-y>

But even if the involvement of M1 CT neurons were restricted to the wheel task, this would not necessarily indicate that they are permissive for learning. Indeed, they could be instead involved in goal-directed motion and/or fine motor control, regardless of the learning status. To show that they are involved specifically in learning, one possibility would be to assess the activity of M1 CT neurons in animals that have mastered the wheel task and for whom there is no learning involved anymore, so when the percentage of successful trials and the maximum velocity have reached a plateau. If the same decrease in activity is observed upon movement as during the late training phase and the same behavioral effect is observed upon optogenetic activation, then it is likely that the neurons are important for the movements underlying the task, rather than the learning of the motor skill.

Taking (1) and (2) together, while the authors are very careful to describe the role of M1 CT neurons as “permissive” for learned motor execution, the authors should provide some additional arguments (data or otherwise) to support a role of the decrease in M1 CT neuronal activity specifically in learning.

Again, we agree with the reviewer that our optogenetic manipulations do not support a role for M1^{CT} neurons in learning and instead are permissive for motor execution. We do not make this claim. We believe the effect of the perturbation increases with training as animals become more proficient at executing the task, but we do not dispute that this could continue well into performance when animals have reached maximal proficiency.

3) The authors control for the differential effect of light across tissue depth by comparing each cluster to its own control. While some clusters are expected to be located within a small depth range (e.g. L2/3 IT), this is not the case for all of them (e.g. PV and SST interneurons are found across layers II-VI), which could lead to an underestimation of the changes in these populations during training. Did the authors control for that particular aspect?

Our approach does not allow us to further control for this. We have analyzed inhibitory cell types, where this is more likely to be a concern, separately from excitatory cell types, but further subdivisions would require new cell type specific CaMPARI tools and histological approaches for further quantitation by location. Because these currently do not exist, we have added this limitation to the interpretation of the data to our “Limitations of the Study” section (lines 551-553).

Additionally, were the control animals run through the entire length of the experiment, and were they time matched to the early and late training animals?

We selected a single control group that would represent untrained animals at the start of the training timeline. The control animals were, thus, initially habituated for the same number of days as trained animals, but they were labeled on what would have been day 1 of training. We have added this information to the text (line 158).

4) While FoxP2-expressing neurons project predominantly to the thalamus, it is not all of them. The authors cleverly address that drawback for their imaging experiments by co-labeling FoxP2 neurons projecting to the spinal cord to restrict their calcium imaging to CT neurons. Did they use a similar approach for their optogenetic experiments, and did they confirm that the majority of the ChrR2-expressing FoxP2 neurons projected to the thalamus? This could otherwise be a confounding factor to assess the role of CT FoxP2 neurons.

By counting the number of corticospinal neurons labeled with the retrograde spinal injection in our calcium imaging mice relative to the number of layer VI corticothalamic neurons, we have also confirmed that layer V corticospinal neurons make up about 11% of FoxP2+ neurons labeled (Rebuttal Figure 11, manuscript S4C).

Rebuttal Figure 11. Percent of all FoxP2 labeled neurons (GCaMP7f expression if FoxP2-cre mice) that are single positive (GCaMP7f, L6) or double positive (GCaMP7f and retrograde, cre-dependent TdTomato injected into cervical spinal cord); L6: 89.2 ± 5.65 , L5: 10.82 ± 5.65 , N=3.

To further address this concern in our optogenetic experiments, we have repeated the manipulation at late learning with light delivered concurrent with trial start in Ntsr1-cre animals. As with our FoxP2 animals, we observed a decrease in the ability of opsin animals to execute the task during light on trials (Rebuttal Figure 12B, C). This is accompanied by a decrease in velocity light on trials in opsin animals but not control animals (Rebuttal Figure 12D, E). These data has also been added to the manuscript (Figure S9).

Rebuttal Figure 12. **A** Schematic of optogenetic light delivery for closed loop at trial start at late training for wheel turning task in *Ntsr1-cre* animals. Light was pulsed at 20 hz; 10 ms pulse width. **B** Percent of successful trials with light on (# successful trials during light on / # of all successful trials); control, gray; hChR2, blue; error bars, SEM; two-tailed unpaired t-test, $p=0.0357$. For B-F: $N=4$, control group; $N=6$, hChR2 group. Each point is average of all light delivery sessions. **C** Percentage of trials that are successful for each animal with light on or off in each group (# successful trials during light on(off) / # of all trials during light on(off)); error bars, SEM; two-way ANOVA, multiple comparisons; $p=0.5302$, control; $p=0.0491$, hChR2, $p=0.3886$ for group x condition. **D** Velocity traces of all trials from control, top, and hChR2, bottom, animals during light, blue, and no light trials, gray; 0=trial start; blue bar denotes time of light, $p>0.9999$ (control), $p<0.0001$ (hChR2), time x condition, two way ANOVA; solid line, mean; shaded area, SEM. **E** Maximum velocity of all trials for each animal with light on or off in each group; two-way ANOVA, multiple comparisons, $p=0.5488$ (control), $p=0.0452$ (hChR2), $p=0.3603$ for group x condition; **F** Wheel distance traveled during all trials for each animal with light on or off for each group; two way ANOVA, multiple comparisons, $p=0.8619$, control, $p=0.0997$, hChR2, $p=0.3203$ for group x condition. Note this figure is the same as Rebuttal Figure 2.

In a similar vein and since the authors did an extensive mapping of the projections of FoxP2 neurons, it would be good to see a quantification of the non-thalamic projections in Fig. S3. Knowing the proportion of these neurons would help assess how large of a confounding factor they could be.

We had only focused these experiments on examining thalamic targets and therefore, cannot quantitate non-thalamic projections. However, previous studies¹. and mapping efforts² (Rebuttal Figure 13) indicate that layer VI neurons do not project outside of the thalamus.

Rebuttal Figure 13. Reconstruction data of all 19 neurons that contain soma in layer 6 of the primary motor cortex and project to thalamus from Janelia's MouseLight project.

The topographic organization of the FoxP2 axons across the thalamus is a particularly interesting piece of data, and following up on which thalamic nuclei are involved would be highly interesting, as the authors suggest. Based on the positioning of their cranial window and other landmarks, could the authors estimate to which thalamic region the neurons that they imaged or inhibited project? This could provide a clue regarding which thalamic nuclei are involved and the interaction between corticothalamic and thalamocortical projections in the circuit.

In our imaging and optogenetic experiments, we have aimed to target the caudal forelimb area (CFA). When putting this in the context of our thalamic projection mapping, we would expect that most of the neurons predominantly target the nuclei that encompass the regions we have defined as lateral and ventral. We have this point to the discussion (lines 524-528).

5) The finding of increased M1 CT neuronal activity during the late-stage non-movement period is quite striking (Figs. 1 and S6), which could contribute to the larger degree of suppression at the late stage of training and may be more directly associated with learning. It would be nice if authors could discuss how increased baseline M1 CT neuronal activities and increased inhibitory inputs onto corticospinal neurons contribute to regulating the learned motor skills (controlling the optimal timing of the action initiation, restricting unwanted/nonspecific movement, etc.) We believe that including this discussion would provide a more comprehensive perspective on the role of M1 CT neurons in this motor learning paradigm.

We have added this point to the discussion (lines 509-513).

Minor concerns

All relevant figures: please add the velocity and distance unit in the figure and figure legend where appropriate.

We have added the units to all plots including velocity (cm/s) and distance (cm). Please note walking during imaging was performed on a disk-like walking wheel. Because the circumference of this wheel is depended on the angle of each animal's body position when walking, we have left these plots (Figure S11C and S12D) as arbitrary units. If the reviewer prefers, we can estimate this to the animal's fixed head position to convert to centimeters.

Fig. S1: Given that UV has limited spread in vivo, could authors provide an accompanying quantitation of the estimated conversion rates across the layers? This information will benefit other researchers who want to adapt the authors' experimental paradigm to their studies.

It is difficult to classify photoconverted neurons because of range of CaMPARI photoconversion. However, we have calculate irradiance at the brain surface to 1mm of cortical depth (Rebuttal Table 2 and Rebuttal Figure 14). The parameters and equations used are also outlined below. We have modified the Methods section to include flux density at the cranial window as well as the table and the outline of our calculations.

cortical depth (um)	Irradiance (mW/mm ²)
100 um	49.282
200 um	29.299
300 um	17.419
400 um	10.356
500 um	6.157
600 um	3.66
700 um	2.176
800 um	1.294
900 um	0.769
1000 um	0.457

Rebuttal Table 2 and Rebuttal Figure 14. Irradiance calculated at the cranial window and at every 100 um of cortical depth until 1mm. The experimental parameters and equations utilized are listed below. Note that Rebuttal Figure 13 is the same as Rebuttal Figure 8.

Experimental parameters:

LED light source output: 60 mW

Light wavelength: 405 nM

Fiber diameter: 960 uM

Fiber NA: 0.63

Irradiance at the fiber tip (E_0):

$$E_0 = \frac{P}{\pi\left(\frac{d}{2}\right)^2}$$

P : Power = 60 mW = 0.06W

d : diameter of fiber = 960 uM = 0.00096 m

$$E_0 = \frac{0.06}{\pi\left(\frac{0.00096}{2}\right)^2}$$

$$E_0 = 82893 \text{ W}/m^2$$

Irradiance at various cortical depths:

$$E(z) = E_0 \cdot e^{-\mu_t z}$$

z : depth

μ_t : total attenuation coefficient

To calculate the total attenuation coefficient (μ_t):

$$\mu_t = \mu_a + \mu_s$$

μ_a : absorption coefficient

estimated to be 0.2 mm^{-1}

μ_s : scattering coefficient

estimated to be between 2 mm^{-1}

estimates from reference:

- 1 Yaroslavsky, A. N. *et al.* Optical properties of selected native and coagulated human brain tissues in vitro in the visible and near infrared spectral range. *Phys Med Biol* **47**, 2059-2073 (2002). <https://doi.org:10.1088/0031-9155/47/12/305>

$$\mu_t = 0.2 + 5$$
$$\mu_t = 5.2 \text{ mm}^{-1}$$

Sample calculation for 100 um below the surface:

$$E(100\text{um}) = E_0 \cdot e^{-\mu_t z}$$
$$E(100\text{um}) = 82893 \cdot e^{-5200(0.0001)}$$
$$E(100\text{um}) = 82893 \cdot e^{-0.52}$$
$$E(100\text{um}) = 82893 \cdot 0.5945$$
$$E(100\text{um}) = 49,281.59$$

Fig. S2A-B: The clusters are presented in a different order between Fig. S2A and S2B, which itself differs from the order in Fig. 1F. Unless there is a particular reason for that different order, we would recommend stating the cluster identity (i1, i2, etc) and rearranging the clusters following the order given in Fig. 1F, to make the comparison between Fig. 1F, S2A and S2B easier. Please also do so for the additional statistics presented in Table 2.

S2A and S2B now match the cluster order of Figure 1G and have cluster ID enumerated.

Fig. S2B: Presumably the M1 cell atlas is displayed on the X axis, but it should be stated explicitly. It is also unclear what the axes refer to (especially as the color key does not cover the full range of colors displayed in the table), so a more detailed description of that panel in the figure legend would be helpful.

We have edited this figured legend to indicate the number of identified clusters on the x axis and the M1 atlas clusters on the Y axis. We apologize for the color range of the legend that appears to have been a consequence of down sampling and have also fixed this.

Figs. 3D and E show a drastic decrease in the maximal velocity for the hChR2 group in both the traces and the paired line plot. However, in Figs S8B and E, when separated by whether the trials were successful, there was no evident reduction in maximal velocity in the traces of both cases. Furthermore, in all cases, the values of paired line plots were higher than those in the traces. Could the authors address what might cause these discrepancies or at least clarify how the velocity measure (and other behavioral measures used in all figures) was quantified in the method section?

The differences in the traces are derived from various sources. First, there is difference in the proportion of successful to unsuccessful trials (Rebuttal Table 1). Because there is a higher proportion of unsuccessful trials, particularly in the opsin group, it will lead to a lower max velocity in the overall trace of all trials. Secondly, the successful and unsuccessful trials do not peak in max velocity at the same time, leading to lower velocities in the overall trace of all trials due to this lack of alignment. Finally, the traces are derived from the average of each animal in each group, while the line graphs measure the peak velocity averaged across trials but per animal. Given inter animal variation, the peak velocity is lower in the trace when averaged across animals with different peak velocities at achieved at different times.

We have added a description of this to our methods section "Optogenetic Manipulations" (lines 868-870).

	on		off	
	unsuccessful	successful	unsuccessful	successful
control	0.742	0.258	0.711	0.289
opsin	0.8782	0.1218	0.677	0.323

Rebuttal Table 3. Proportion of unsuccessful and successful trials executed with light on or off for control and opsin groups. Note that Rebuttal Table 3 is the same as Rebuttal Table 1.

Fig. S9C/F: Three animals show a negative distance, including in a successful trial. Did the animals push the wheel instead of pulling it? Shouldn't they be excluded in that case, since that movement wasn't part of training?

We chose not to exclude these animals because they still met all the necessary contingencies for a trial. While they push the wheel for some portion of the trial, they also pull during this time period (see example trial, Rebuttal Figure 15). We think it most objective to not remove specific animals or trials post-hoc as the animals did not train with these contingencies, and this could bias towards specific execution strategies which we do not intend.

Rebuttal Figure 15. Example successful trial where mouse pulls wheel with sufficient velocity for reward but also pulls wheel leading so negative distances and velocities.

Line 171: Please include the references to the published annotations.

We apologize for the lack of clarity. The annotations had been referenced earlier in the sentence, but this has now been moved to the end of the sentence.

Lines 183-185: please specify in the text which cluster numbers the authors referred to (see line 189) for readers to better cross-reference Figure 1.

We have added the corresponding cluster ID to all comparisons.

Line 266: "Decrease" should read "decreased" or "decrease in".

This has been changed to "decreased".

Line 595: "Later" should read "lateral".

This has been fixed.

Lines 599 – 610: The units for the virus volumes are inconsistent. mL and, in some places, uL seem to be used by error throughout the section. Please double-check and correct the typos accordingly.

Thank you for the careful read of the methods! We have fixed these.

We thank the reviewers for considering the additions to our manuscript. We are pleased that our revisions have addressed all concerns from reviewers 2 and 3, and all major concerns from reviewer 1. We appreciate the additional feedback on this version of the manuscript and have made further additions or edits to address these comments. We have modified 8 figure panels throughout as outlined below, and we have made several edits to the manuscript (highlighted in yellow). We have also included a more thorough table which includes information on statistical tests and values reported for each panel of each figure.

We hope these changes have addressed all remaining concerns and now make the manuscript suitable for publication in Nature Communications. Please find below a point-by-point answer to each reviewer's comments (reviewer comment in black and our response in blue).

REVIEWER COMMENTS

Reviewer #1 (Remarks to the Author):

The revisions satisfactorily address most of the issues. The quantification of FoxP2 spillover into corticospinals, replication of the behavioral experiments in Ntsr1-cre mice, quantification of kinematics, and analysis of excitation and inhibition latencies strengthen the conclusions of this thorough and intriguing study. The newly included supplementary movies are informative. There are a few remaining concerns, primarily about clarification of the new experiments and analyses.

It is not clear how exactly were the data shuffled for the shuffling analysis. Were fluorescence traces completely scrambled, or were they shuffled in such a way as to preserve the autocorrelation structure while breaking the temporal relation to wheel velocity, e.g. trial shuffling or session shuffling? How many shuffling iterations were performed?

For panels S4D, S4G, and S13B (also below; Rebuttal Figure 1B, 1E and 1H), the fluorescent traces were completely scrambled over the whole session to match the cross-correlation analysis performed over the whole session in panels 2C, 2F and 5B, respectively (also below; Rebuttal Figure 1A, 1D, and 1G). For panel S13F (also below; Rebuttal Figure 1K), we extracted the signal from each trial and shuffled this before calculating the cross-correlation to the wheel velocity. This is the same as was done with the unshuffled signal for panel 5C, (also below; Rebuttal Figure 1J)

We had previously performed a single shuffling iteration as described above. We have now performed 100 iterations and averaged the resulting cross-correlation over these iterations. The corresponding panels in the manuscript have been updated (S14D, S4G, S13B, and S13F; also below: Rebuttal Figure 1C, F, I, L), and we have attempted to clarify the shuffling strategy in each legend.

Rebuttal Figure 1. **A** Cross-correlation of Z-scored calcium $\Delta F/F$ for each unit of corticothalamic neurons in FoxP2 animals to wheel velocity for whole session; mean, solid gray line; shaded area, SEM; **B,C** Cross-correlation of shuffled Z-scored calcium $\Delta F/F$ for each unit of corticothalamic neurons in FoxP2 animals to wheel velocity for whole session; mean, solid gray line; shaded area. **B**, one shuffling iteration; **C**, average of 100 shuffling iterations. **D** Cross-correlation of Z-scored calcium $\Delta F/F$ to wheel velocity for whole session for each group of corticothalamic neurons from FoxP2-cre animals; solid lines, mean; shaded areas, SEM; wheel velocity during pulls. **E,F** Cross-correlation for same groups as in **D** but with shuffled Z-scored calcium $\Delta F/F$; **E**, one shuffling iteration; **F**, average of 100 shuffling iterations. **G** Cross-correlation of Z-scored calcium $\Delta F/F$ for each unit of corticospinal neurons to wheel velocity for whole session; mean, solid gray line; shaded area, SEM. **H,I** Cross-correlation of shuffled Z-scored calcium $\Delta F/F$ for each unit of corticospinal neurons to wheel velocity for whole session; mean, solid gray line; shaded area. **H**, one shuffling iteration; **I**, average of 100 shuffling iterations. **J** Cross-correlation of Z-scored calcium $\Delta F/F$ for corticospinal neurons to wheel velocity for pulls at early (blue) and late (green) training sessions; solid line, mean; shaded area, SEM. **K,L** Cross-correlation of shuffled Z-scored calcium $\Delta F/F$ for corticospinal neurons to wheel velocity for wheel pulls; mean, solid gray line; shaded area; **K**, one shuffling iteration; **L**, average of 100 shuffling iterations.

The inclusion of descriptive statistics for each group in each ANOVA is excessive and hampers readability. Move these to the tables showing the p-values of pairwise comparisons, and report descriptive statistics in the text only for two-group tests (e.g. t-tests). These tables should also include standard ANOVA summary tables (degrees of freedom, sums of squares, etc. for each effect tested and residual error, plus F statistics and p-values for effects), and should also specify which effects were modelled as within or between subjects. Finally, almost all numbers in the text are reported to an excessive number of significant figures. Two to three is enough in most cases.

We have made a single table (manuscript Table 3; also attached here) that now includes all the statistical test information by figure panel as well as descriptive statistics and N/n values. We have also reduced reported values throughout the manuscript to 3 significant figures or 3 values after the decimal.

Regarding my previous point about stimulation before movement initiation, reporting percent success for category 1 and category 2 trials from Rebuttal Figure 6 would address this concern, even if the number of trials is low. It can still be calculated if there are more than zero trials, which is the case for most mice. It would be fine to pool these two categories to help increase the trial numbers.

We apologize if prior rebuttal figure 6 (also below as Rebuttal Figure 2) was confusing. The percents presented in each category are of all trials not only successful. Thus, the number of successful trials within these is indeed zero for most animals. We have aggregated category 1 and 2 and computed percent of success (# successful trials during light on / # of all successful trials) and percentage of trials that are successful for each animal with light on or off in each group (# successful trials during light on (OR off) / # of all trials with light on (OR off)). They are as follows:

Percent of success:

control 0.692± 0.998,
hChR2 0.181± 0.412
two-tailed unpaired t-test, p=0.135

Percentage success:

	Light off	Light on
control	29.526 ± 11.012	20.000 ± 25.820
hChR2	36.377 ± 15.730	13.636 ± 32.333

two-way ANOVA, multiple comparisons; p=0.502, control; p=0.0282, hChR2, p=0.293 for group x condition

However, we believe it would be statistically misleading to report these and have not included them in the manuscript. Furthermore, we would like to again note that while we cannot rule out an effect on success, we do not observe an effect on trial initiation.

Rebuttal Figure 2. Percent of all trials, as depicted in each category (S, green, trial start; R, red, trial end, blue bar, light on period), that fall within a period of light on or a subset of light off periods (subsampling to match total time) for control or hChR2 mice.

Minor comments:

More raw data should be shown for the quantification of kinematics, e.g. average traces similar to those for wheel velocity, before jumping to summary statistics.

As previously noted, we computed max distance and total distance in our kinematic analysis because the low sampling rate of our videos do not allow for trajectory analysis. However, we agree that raw data could be informative and have included density plots of the position of the forelimb at max distance for each mouse for light on and light off trials (Rebuttal Figure 3). As in our summary data, it is visually noticeable that hChR2 animals have distribution closer to the start point of the trial (origin) in light on trials.

Rebuttal Figure 3. Density plots of right forelimb position at maximal distance from the position at start of each trial for each mouse during light off or on trials; values offset to start position (origin).

Where percentage of successful trials that had light on is reported, the text changes are helpful, but in the figures (Figs 3B,H; S9B; S11B), a change of y-axis label from “% of success” to e.g. “% light on (successful trials)” would make things even clearer.

We agree and have changed the relevant y-axis labels accordingly.

Line 305: should read “decrease of the maximum distance traveled by the right forelimb”.

This has been changed accordingly.

Line 434: should read “stimulation of M1CT evoked relatively small excitatory postsynaptic currents”

This has been changed accordingly.

Fig S3: the A/P profiles of retrograde cortical labeling from thalamus in Fig S3F do not seem to match the 2D profiles in Fig S3G. Are the medial and lateral labels flipped in Fig S3G?

Thank you for catching this. The labels were indeed swapped and have been amended.

Fig S4: the quantification in Fig S4C is useful, but looking at the example in Fig S4B, there appear to be one or two GFP+, tdTomato -ve cells in layer 5 (based on having similar depth to tdTomato +ve corticospinal cells). If the layer boundaries can be accurately determined from the bright field images of these slices, an even better quantification would be L6 GFP+ vs L5 GFP+/tdTom+ vs L5 GFP+/tdTom-, to estimate the proportion of non-retrogradely-labelled FoxP2 +ve corticospinals that could be contaminating the fluorescence data.

We have attempted to subdivide the labeled layer V neurons into GFP+/tdTom+, GFP+/tdTom-, and GFP-/tdTom+ (Rebuttal Figure 4). However, it is important to note that these are estimates based on depth and labeling within each slice. While we can confidently identify any neuron expressing tdTomato as a corticospinal, neurons expressing GFP could be from either layer V or VI. We have also updated the corresponding panel (Figure S4C) of the manuscript.

Rebuttal Figure 4. Percent of all FoxP2 labeled neurons that are single positive (GCaMP7f, layer VI, 87.5 ± 5.73), double positive (GCaMP7f and retrograde, cre-dependent TdTomato injected into cervical spinal cord, layer V, 10.8 ± 6.05); single positive (GCaMP7f, layer V, estimated, 1.56 ± 0.516); single positive (cre-dependent TdTomato injected into cervical spinal cord, layer V, 0.156 ± 0.0784); N=3.

Fig S14: panels K and L should use non parametric statistics (i.e. median and median absolute deviation or IQR) due to the highly skew distributions evident in panels I and J. Calculating EPSC to IPSC latency differences per-cell before averaging would also help by accounting for cell-to-cell variability.

We have edited these panels to include all data points and non-parametric statistics (Rebuttal Figure 5; Manuscript Figure S14K,L). We have also added a panel to include the difference in IPSC and EPSC latencies calculated by cell (Rebuttal Figure XX; Manuscript Figure S14M).

Rebuttal Figure 5. **A,B** Quantification of latency of excitatory and inhibitory responses post stimulus. **A**, untrained; **B**, trained; horizontal line, median; vertical line, IQR. **C** Difference in inhibitory and excitatory latencies (IPSC - EPSC per cell) in untrained, gray, or trained, green, animals; horizontal line, median; vertical line, IQR

Reviewer #2 (Remarks to the Author):

The authors have fully addressed our concerns and we support the publication of this excellent study.

Thank you!

Reviewer #3 (Remarks to the Author):

See below.

Thank you!

Table 3				
Figure	sample size	summary statistics	statistical test	values
1B	N=6		one way repeated measures ANOVA (time)	P=0.0104; SS=7664; DF=17; MS=450.8; F(3.623, 18.11)=4.69
	N=6 (day 4); N=6 (day 12)	mean (day 4)=20.88; mean (day 12)=41.22; mean diff.=-20.45; SE of diff=7.039	one way ANOVA; planned comparison between day 4 and day 12; Sidak's multiple comparisons	adjusted P value=0.0336; t=2.904; DF=5
1C	N=6		one way repeated measures ANOVA (time)	P=0.0182; SS=66.82; DF=17; MS=3.93; F(3.422, 17.11)=4.2
	N=6 (day 4); N=6 (day 12)	mean (day 4)=4.069; mean (day 12)=5.806; mean diff.=-1.737; SE of diff=0.747	one way ANOVA; planned comparison between day 4 and day 12; Sidak's multiple comparisons	adjusted P value=0.0569; t=2.465; DF=5
1G	see Table 3 and Methods for description of test and values			
1H	see Table 3 and Methods for description of test and values			
2I	N=4; n=740	2 to 6: mean=-0.120, SD=0.082; 6 to 10: mean=-0.165, SD=0.086; 10 to 14: mean=-0.193, SD=0.104; 14 and up: mean=-0.207, SD=0.124	ordinary one way ANOVA; Tukey's multiple comparison test for all comparisons	P<0.001; SS=3.286; DF=3; MS=1.095; F(3, 2956)=108.4; adjusted P values for multiple comparisons: 2 to 6 vs. 6 to 10: P<0.0001; 2 to 6 vs. 10 to 14: P<0.0001; 2 to 6 vs. 14 and up: P<0.0001; 6 to 10 vs. 10 to 14: P<0.0001; 6 to 10 vs. 14 and up: P<0.0001; 10 to 14 vs. 14 and up: P=0.0553
2J	N=4; n=740	2 to 6: mean=0.063, SD=0.037; 6 to 10: mean=0.091, SD=0.053; 10 to 14: mean=0.099, SD=0.062; 14 and up: mean=0.099, SD=0.061	ordinary one way ANOVA; Tukey's multiple comparison test for all comparisons	P<0.0001; SS=0.663; DF=3; MS=0.221; F(3, 2956)=76.49; adjusted P values for multiple comparisons: 2 to 6 vs. 6 to 10: P<0.0001; 2 to 6 vs. 10 to 14: P<0.0001; 2 to 6 vs. 14 and up: P<0.0001; 6 to 10 vs. 10 to 14: P<0.0001; 6 to 10 vs. 14 and up: P<0.0482; 10 to 14 vs. 14 and up: P=0.994
2M	N=4; n=231 (early), n=228 (late)	2 to 6 early: mean=-0.055, SD=0.041; 2 to 6 late: mean=-0.096, SD=0.058; 6 to 10 early: mean=-0.047, SD=0.104; 6 to 10 late: mean=-0.136, SD=0.079; 10 to 14 early: mean=-0.139, SD=0.053; 10 to 14 late: mean=-0.168, SD=0.098; 14 and up early: mean=-0.161, SD=0.069; 14 and up late: mean=-0.2, SD=0.104	ordinary one way ANOVA; Tukey's multiple comparison test for all comparisons	P<0.0001; SS=3.568; DF=7; MS=0.51; F(7,1828)=98.17; adjusted P values for multiple comparisons: P<0.0001 for all comparisons except the following; 2 to 6 early vs. 6 to 10 early: P<0.999; 6 to 10 late vs. 10 to 14 early: P=0.999; 6 to 10 late vs 14 and up early: P=0.005; 10 to 14 early vs. 10 to 14 late: P=0.0005; 10 to 14 early vs. 14 and up early: P=0.0318; 10 to 14 late vs. 14 and up early, P=0.9582
3B	N=7 control; N=6 hChR2	control, mean=33.8, SD=8.77; hChR2, mean=14.3, SD=14.4	two-tailed unpaired t-test	P=0.0119; t=3.009
3C	N=7 control; N=6 hChR2	control, OFF: mean=28.9; SD=15.754; control, ON: mean=25.8; SD=15.864; hChR2, OFF: mean=32.299; SD=9.429; hChR2, ON: mean=12.184; SD=9.898	two way repeated measures ANOVA; planned comparison between light on and light off for each group; Sidak's multiple comparisons	P=0.0148; SS=467.7; DF=1; MS=467.7; F(1,11)=8.336; adjusted P values for multiple comparisons: control, P=0.703; hChR2, P=0.0014
3D	N=7 control; N=6 hChR2		two way repeated measures ANOVA for each group	control: P>0.999; SS=21.27; DF=799; MS=0.0266; F(799, 9588)=0.0776; hChR2: P<0.0001; SS=540.1; DF=799; MS=0.676; F(799, 7990)=4.28
3E	N=7 control; N=6 hChR2	control, OFF: mean=6.239; SD=2.606; control, ON: mean=5.993; SD=2.630; hChR2, OFF: mean=6.578; SD=1.135; hChR2, ON: mean=3.404; SD=1.424	two way repeated measures ANOVA; planned comparison between light on and light off for each group; Sidak's multiple comparisons	P=0.0026; SS=13.85, DF=1; MS=13.85; F(1, 11)=14.97; adjusted P values for multiple comparisons: control, P=0.871; hChR2, P=0.0003
3F	N=7 control; N=6 hChR2	control, OFF: mean=1359.721; SD=1295.942; control, ON: mean=1300.71; SD=1185.064; hChR2, OFF: mean=1365.124; SD=563.159; hChR2, ON: mean=423.689; SD=432.615	two way repeated measures ANOVA; planned comparison between light on and light off for each group; Sidak's multiple comparisons	P=0.0139; SS=1257857, DF=1; MS=1257857; F(1, 11)=8.545; adjusted P values for multiple comparisons: control, P=0.951; hChR2, P=0.0027

3H	N=5 control; N=6 hChr2	control, mean=37.73, SD=12.51; hChr2, mean=21.94, SD=14.93	two-tailed unpaired t-test	P=0.0933; t=1.876
3I	N=5 control; N=6 hChr2	control, OFF: mean=18.025; SD=14.025; control, ON: mean=19.182; SD=11.329; hChr2, OFF: mean=15.051; SD=7.328; hChr2, ON: mean=8.841; SD=9.519	two way repeated measures ANOVA; planned comparison between light on and light off for each group; Sidak's multiple comparisons	P=0.133; SS=63.19; DF=1; MS=63.19; F(1,9)=2.724; adjusted P values for multiple comparisons: control, P=0.977; hChr2, P=0.102
3J	N=5 control; N=6 hChr2		two way repeated measures ANOVA for each group	control: P>0.999; SS=108.7; DF=799; MS=0.136; F(799, 6392)=0.686; hChr2: P<0.0001; SS=225.9; DF=799; MS=0.283; F(799, 7990)=2.356
3K	N=5 control; N=6 hChr2	control, OFF: mean=5.718; SD=2.293; control, ON: mean=5.778; SD=2.155; hChr2, OFF: mean=4.823; SD=1.288; hChr2, ON: mean=3.922; SD=1.32	two way repeated measures ANOVA; planned comparison between light on and light off for each group; Sidak's multiple comparisons	P=0.0642; SS=1.26; DF=1; MS=1.26; F(1,9)=4.448; adjusted P values for multiple comparisons: control, P=0.981; hChr2, P=0.0331
3L	N=5 control; N=6 hChr2	control, OFF: mean=1059.868; SD=948.379; control, ON: mean=1214.159; SD=868.175; hChr2, OFF: mean=798.553; SD=575.544; hChr2, ON: mean=530.696; SD=472.81	two way repeated measures ANOVA; planned comparison between light on and light off for each group; Sidak's multiple comparisons	P=0.0837; SS=243012; DF=1; MS=243012; F(1,9)=3.78; adjusted P values for multiple comparisons: control, P=0.592; hChr2, P=0.191
4B	N=6 control; N=6 hChr2		two way repeated measures ANOVA for each group	control: P=0.862; SS=126.4; DF=799; MS=0.158; F(799, 7990)=0.943; hChr2: P=0.0003; SS=344.7; DF=799; MS=0.431; F(799, 7990)=1.190
4C	N=6 control; N=6 hChr2	control, OFF: mean=7.001; SD=1.843; control, ON: mean=6.657; SD=1.424; hChr2, OFF: mean=6.301; SD=2.357; hChr2, ON: mean=5.187; SD=2.078	two way repeated measures ANOVA; planned comparison between light on and light off for each group; Sidak's multiple comparisons	P=0.2766; SS=0.887; DF=1; MS=0.887; F(1,10)=1.324; adjusted P values for multiple comparisons: control, P=0.733; hChr2, P=0.0789
4D	N=6 control; N=6 hChr2	control, OFF: mean=1910.372; SD=874.505; control, ON: mean=1633.975; SD=692.672; hChr2, OFF: mean=1497.517; SD=1202.637; hChr2, ON: mean=799.206; SD=807.669	two way repeated measures ANOVA; planned comparison between light on and light off for each group; Sidak's multiple comparisons	P=0.2674; SS=267017; DF=1; MS=267017; F(1,10)=1.38; adjusted P values for multiple comparisons: control, P=0.513; hChr2, P=0.0406
4F	N=7 control; N=7 hChr2		two way repeated measures ANOVA for each group	control: P>0.999; SS=1401; DF=39; MS=35.93; F(39, 468)=0.0277; hChr2: P=0.996; SS=27784; DF=39; MS=712.4; F(39, 546)=0.497
4G	N=7 control; N=7 hChr2	control, OFF: mean=6.217; SD=0.598; control, ON: mean=5.982; SD=0.685; hChr2, OFF: mean=6.345; SD=0.562; hChr2, ON: mean=6.534; SD=1.059	two way repeated measures ANOVA; planned comparison between light on and light off for each group; Sidak's multiple comparisons	P=0.324; SS=0.336; DF=1; MS=0.336; F(1,13)=1.051; adjusted P values for multiple comparisons: control, P=0.698; hChr2, P=0.765
4H	N=7 control; N=7 hChr2	control, OFF: mean=1.548; SD=0.329; control, ON: mean=1.472; SD=0.402; hChr2, OFF: mean=1.611; SD=0.204; hChr2, ON: mean=1.641; SD=0.408	two way repeated measures ANOVA; planned comparison between light on and light off for each group; Sidak's multiple comparisons	P=0.546; SS=0.021; DF=1; MS=0.021; F(1,13)=0.385; adjusted P values for multiple comparisons: control, P=0.8; hChr2, P=0.961
5H	N=2, n=22, untrained; N=2, n=24, trained	inhibitory, untrained: mean=79.431; SD=129.675; inhibitory, trained: mean=272.822; SD=238.078; excitatory, untrained: mean=38.464; SD=52.596; excitatory, trained: mean=48.566; SD=0.44.660	two way repeated measures ANOVA; Sidak's multiple comparisons	P=0.0026; SS(Type II)=192805; DF=1; MS=192805; F(1, 88)=9.636; adjusted P values for multiple comparisons: untrained excitatory vs. untrained inhibitory: P=0.917; untrained excitatory vs trained excitatory: P>0.999; untrained excitatory vs. trained inhibitory: P<0.0001; untrained inhibitory vs. trained excitatory: P=0.976; untrained inhibitory vs. trained inhibitory: P<0.0001; trained excitatory vs. trained inhibitory: P<0.0001
5I	N=2, n=22, untrained; N=2, n=24, trained	untrained: mean=5.371; SD=3.35; trained: mean=2.91; SD=3.122	two-tailed unpaired t-test	P=0.0136; t=2.571

S3D	N=4	layer V, lateral: mean=2248; SD=1463.277; medial: mean=4293.25; SD=2436.037; ventral: mean=7379.25; SD=3668.223; layer VI, lateral: mean=10029; SD=2314.81; medial: mean=13150.75; SD=3625.901; ventral: mean=19393.5; SD=4730.042	one way repeated measures ANOVA; Tukey's multiple comparisons for each layer independently	layer V: P=0.018; SS=53381560; DF=2; MS=26690780; F(1.763, 5.290)=9.747; adjusted P values for multiple comparisons, lateral vs. ventral, P=0.0513; lateral vs. medial, P=0.22; ventral vs. medial, P=0.188; layer VI: P=0.0156; SS=181881481; DF=2; MS=90940741; F(1.847, 5.542)=9.867; adjusted P values for multiple comparisons, lateral vs. ventral, P=0.0516; lateral vs. medial, P=0.333; ventral vs. medial, P=0.144
S3F	N=4	layer V anterior, lateral: mean=253.35; SD=237.666; ventral: mean=2778; SD=848.012; medial: mean=3019.5; SD=1836.014; layer V posterior, lateral: mean=1272.25; SD=1025.193; ventral: mean=2827.75; SD=2261.514; medial: mean=482.5; SD=288.437; layer VI anterior, lateral: mean=1462; SD=1149.387; ventral: mean=8351.5; SD=2076.224; medial: mean=9011.75; SD=1998.932; layer VI posterior, lateral: mean=5636; SD=684.726; ventral: mean=6887.75; SD=1827.258; medial: mean=2270.5; SD=993.550	two way ordinary ANOVA; Tukey's multiple comparisons	adjusted P values for multiple comparisons, layer V anterior: lateral vs. ventral, P<0.0001; lateral vs. medial, P<0.0001; ventral vs. medial, P=0.821; layer V posterior: lateral vs. ventral, P=0.502; lateral vs. medial, P=0.017; ventral vs. medial, P=0.0014; layer VI anterior: lateral vs. ventral, P=0.0362; lateral vs. medial, P=0.0212; ventral vs. medial, P=0.964; layer VI posterior: lateral vs. ventral, P=0.243; lateral vs. medial, P=0.679; ventral vs. medial, P=0.0533;
S5A	N=4; n=740	2 to 6: mean=2139; SD=147.8; 6 to 10: mean=2168, SD=140; 10 to 14: mean=2171; SD=145.4; 14 and up: mean=2175; SD=145.1	two way repeated measures ANOVA; Tukey's multiple comparisons	P<0.0001; SS=613778; DF=3; MS=204593; F(3, 2956)=9.783; adjusted P values for multiple comparisons: 2 to 6 vs 6 to 10, P=0.0004; 2 to 6 vs. 10 to 14, P=0.0001; 2 to 6 vs. 14 and up, P<0.0001; 6 to 10 vs. 10 to 14, P=0.996; 6 to 10 vs. 14 and up, P=0.832; 10 to 14 vs 14 and up, P=0.924
S5B	N=4; n=740	2 to 6: mean=-0.0504; SD=0.0344; 6 to 10: mean=-0.0552, SD=0.0359; 10 to 14: mean=-0.06; SD=0.0384; 14 and up: mean=-0.067; SD=0.0404	two way repeated measures ANOVA; Tukey's multiple comparisons	P<0.0001; SS=0.1049; DF=3; MS=0.035; F(3, 2956)=25.1; adjusted P values for multiple comparisons: 2 to 6 vs 6 to 10, P=0.0711; 2 to 6 vs. 10 to 14, P<0.0001; 2 to 6 vs. 14 and up, P<0.0001; 6 to 10 vs. 10 to 14, P=0.0657; 6 to 10 vs. 14 and up, P<0.0001; 10 to 14 vs 14 and up, P=0.0039
S6C	trial off: N=4; trial on: N=4	trial off: mean=26.352, SD=8.247; trial on: mean=7.973, SD=3.836	two-tailed unpaired t-test	P=0.0068; t=4.041
S6D	N=4	early: mean=0.196, SD=0.0946; late: mean=0.223, SD=0.111	two-tailed unpaired t-test	P<0.001; t=5.66
S7B	N=4; 2 to 6 early, n=8910; 2 to 6 late, n=11150; 6 to 10 early, n=2451; 6 to 10 late, n=3555; 10 to 12 early, n=1235; 10 to 14 late, n=2161; 14 and up early, n=1240; 14 and up late, n=2168	2 to 6 early: mean=3.608, SD=1.136; 2 to 6 late: mean=3.583, SD=1.153; 6 to 10 early: mean=7.68, SD=1.138; 6 to 10 late: mean=7.699, SD=1.155; 10 to 14 early: mean=11.82, SD=1.154; 10 to 14 late: mean=11.84, SD=1.17; 14 and up early: mean=20.16, SD=5.172; 14 and up late: mean=20.76, SD=5.634	ordinary one way ANOVA; Tukey's multiple comparison test for all comparisons	P<0.0001; SS=939946; DF=7; MS=134278; F(7,32862)=31349; adjusted P values for multiple comparisons: P<0.0001 for all comparisons except the following; 2 to 6 early vs. 2 to 6 late: P=0.989; 6 to 10 early vs. 6 to 10 late: P>0.999; 10 to 14 early vs 10 to 14 late: P>0.999;
S8B	N=7 control; N=6 hChr2		two way repeated measures ANOVA for each group	control: P>0.999; SS=118; DF=799; MS=0.148; F(799, 9588)=0.0982; hChr2: P<0.001; SS=4335; DF=799; MS=5.426; F(799, 7191)=2.99
S8C	N=7 control; N=6 hChr2	control, OFF: mean=10.356; SD=2.502; control, ON: mean=10.181; SD=2.972; hChr2, OFF: mean=11.299; SD=1.107; hChr2, ON: mean=9.477; SD=1.468	two way repeated measures ANOVA; planned comparison between light on and light off for each group; Sidak's multiple comparisons	P=0.0327; SS=3.951, DF=1; MS=3.951; F(1, 10)=6.14; adjusted P values for multiple comparisons: control, P=0.905; hChr2, P=0.0098

S8D	N=7 control; N=6 hChR2	control, OFF: mean=2743.694; SD=1696.027; control, ON: mean=2592.003; SD=1790.459; hChR2, OFF: mean=3174.612; SD=406.003; hChR2, ON: mean=1931.767; SD=1132.402	two way repeated measures ANOVA; planned comparison between light on and light off for each group; Sidak's multiple comparisons	P=0.0451; SS=1736319, DF=1; MS=1736319; F(1, 10)=5.241; adjusted P values for multiple comparisons: control, P=0.865; hChR2, P=0.0132
S8E	N=7 control; N=6 hChR2		two way repeated measures ANOVA for each group	control: P>0.999; SS=24.19; DF=799; MS=0.0303; F(799, 9588)=0.0146; hChR2: P<0.001; SS=266.7; DF=799; MS=0.334; F(799, 7990)=3.301
S8F	N=7 control; N=6 hChR2	control, OFF: mean=4.198; SD=1; control, ON: mean=4.199; SD=1.013; hChR2, OFF: mean=4.411; SD=0.525; hChR2, ON: mean=2.635; SD=0.846	two way repeated measures ANOVA; planned comparison between light on and light off for each group; Sidak's multiple comparisons	P=0.004; SS=5.105, DF=1; MS=5.105; F(1, 11)=12.78; adjusted P values for multiple comparisons: control, P>0.999; hChR2, P=0.001
S8G	N=7 control; N=6 hChR2	control, OFF: mean=516.414; SD=318.142; control, ON: mean=592.114; SD=334.414; hChR2, OFF: mean=582.893; SD=292.127; hChR2, ON: mean=196.686; SD=193.102	two way repeated measures ANOVA; planned comparison between light on and light off for each group; Sidak's multiple comparisons	P=0.0149; SS=344654, DF=1; MS=344654; F(1, 11)=8.303; adjusted P values for multiple comparisons: control, P=0.751; hChR2, P=0.0145
S8H	N=7 control; N=6 hChR2	control, OFF: mean=47.215; SD=13.313; control, ON: mean=39.973; SD=12.779; hChR2, OFF: mean=51.820; SD=13.728; hChR2, ON: mean=36.125; SD=17.423	two way repeated measures ANOVA; planned comparison between light on and light off for each group; Sidak's multiple comparisons	P=0.186; SS=115.4; DF=1; MS=115.4; F(1, 11)=1.99; adjusted P values for multiple comparisons: control, P=0.195; hChR2, P=0.0088
S9B	N=4 control; N=6 hChR2	control, mean=34.868, SD=4.616; hChR2, mean=28.877, SD=2.984	two-tailed unpaired t-test	P=0.0357; t=2.521
S9C	N=7 control; N=6 hChR2	control, OFF: mean=33.545; SD=21.723; control, ON: mean=29.894; SD=21.536; hChR2, OFF: mean=34.392; SD=9.398; hChR2, ON: mean=26.736; SD=5.261	two way repeated measures ANOVA; planned comparison between light on and light off for each group; Sidak's multiple comparisons	P=0.389; SS=19.25; DF=1; MS=19.25; F(1, 8)=0.831; adjusted P values for multiple comparisons: control, P=0.53; hChR2, P=0.049
S9D	N=7 control; N=6 hChR2		two way repeated measures ANOVA for each group	control: P>0.999; SS=67.55; DF=799; MS=0.0846; F(799, 4794)=0.341; hChR2: P<0.001; SS=290.8; DF=799; MS=0.364; F(799, 7990)=1.274
S9E	N=7 control; N=6 hChR2	control, OFF: mean=4.713; SD=1.333; control, ON: mean=4.236; SD=1.375; hChR2, OFF: mean=5.729; SD=1.174; hChR2, ON: mean=4.679; SD=0.416	two way repeated measures ANOVA; planned comparison between light on and light off for each group; Sidak's multiple comparisons	P=0.36; SS=0.395, DF=1; MS=0.395; F(1, 8)=0.941; adjusted P values for multiple comparisons: control, P=0.549; hChR2, P=0.0452
S9F	N=7 control; N=6 hChR2	control, OFF: mean=530.737; SD=234.22; control, ON: mean=478.182; SD=206.218; hChR2, OFF: mean=734.782; SD=289.356; hChR2, ON: mean=548.276; SD=101.972	two way repeated measures ANOVA; planned comparison between light on and light off for each group; Sidak's multiple comparisons	P=0.36; SS=24488, DF=1; MS=24488; F(1, 8)=1.122; adjusted P values for multiple comparisons: control, P=0.862; hChR2, P=0.0997
S10A	N=5 control; N=5 hChR2		two way repeated measures ANOVA for each group	control: P>0.999; SS=1815; DF=799; MS=2.272; F(799, 6392)=0.723; hChR2: P>0.999; SS=1619; DF=799; MS=2.027; F(799, 6392)=0.729
S10B	N=5 control; N=5 hChR2	control, OFF: mean=9.939; SD=2.136; control, ON: mean=9.992; SD=1.914; hChR2, OFF: mean=8.874; SD=0.899; hChR2, ON: mean=9.532; SD=1.67	two way repeated measures ANOVA; planned comparison between light on and light off for each group; Sidak's multiple comparisons	P=0.52; SS=0.457, DF=1; MS=0.457; F(1, 8)=0.454; adjusted P values for multiple comparisons: control, P=0.996; hChR2, P=0.552
S10C	N=5 control; N=5 hChR2	control, OFF: mean=2272.762; SD=1815.832; control, ON: mean=2018.416; SD=3239.149; hChR2, OFF: mean=2219.905; SD=902.005; hChR2, ON: mean=2010.021; SD=854.026	two way repeated measures ANOVA; planned comparison between light on and light off for each group; Sidak's multiple comparisons	P=0.96; SS=2471, DF=1; MS=2471; F(1, 8)=0.00264; adjusted P values for multiple comparisons: control, P=0.903; hChR2, P=0.933

S10D	N=5 control; N=6 hChR2		two way repeated measures ANOVA for each group	control: P>0.999; SS=35.75; DF=799; MS=0.0447; F(799, 6392)=0.265; hChR2: P<0.001; SS=192.5; DF=799; MS=0.241; F(799, 7990)=2.583
S10E	N=5 control; N=6 hChR2	control, OFF: mean=4.657; SD=1.529; control, ON: mean=4.793; SD=1.524; hChR2, OFF: mean=4.156; SD=1.157; hChR2, ON: mean=3.471; SD=0.906	two way repeated measures ANOVA; planned comparison between light on and light off for each group; Sidak's multiple comparisons	P=0.0405; SS=0.92, DF=1; MS=0.92; F(1, 9)=5.718; adjusted P values for multiple comparisons: control, P=0.844; hChR2, P=0.0316
S10F	N=5 control; N=6 hChR2	control, OFF: mean=624.243; SD=651.045; control, ON: mean=760.872; SD=550.083; hChR2, OFF: mean=506.562; SD=452.633; hChR2, ON: mean=384.972; SD=345.679	two way repeated measures ANOVA; planned comparison between light on and light off for each group; Sidak's multiple comparisons	P=0.092; SS=90922, DF=1; MS=90922; F(1, 9)=0.092; adjusted P values for multiple comparisons: control, P=0.376; hChR2, P=0.393
S11A	N=6 control; N=6 hChR2		two way repeated measures ANOVA for each group	control: P=0.862; SS=126.4; DF=799; MS=0.158; F(799, 7990)=0.943; hChR2: P=0.0003; SS=344.7; DF=799; MS=0.431; F(799, 7990)=1.190
S11B	N=6 control; N=6 hChR2	control, mean=32.12, SD=7.24; hChR2, mean=24.47, SD=10.2	two-tailed unpaired t-test	P=0.164; t=1.5
S11C	N=6 control; N=6 hChR2	control, OFF: mean=35.745; SD=14.184; control, ON: mean=33.729; SD=14.296; hChR2, OFF: mean=39.483; SD=18.945; hChR2, ON: mean=32.078; SD=17.742	two way repeated measures ANOVA; planned comparison between light on and light off for each group; Sidak's multiple comparisons	P=0.092; SS=90922, DF=1; MS=90922; F(1, 9)=0.092; adjusted P values for multiple comparisons: control, P=0.376; hChR2, P=0.393
S11D	N=6 control; N=6 hChR2		two way repeated measures ANOVA for each group	control: P=0.999; SS=455.2; DF=799; MS=0.57; F(799, 7990)=0.852; hChR2: P<0.0001; SS=2210; DF=799; MS=2.766; F(799, 7990)=3.023
S11E	N=6 control; N=6 hChR2	control, OFF: mean=10.226; SD=2.09; control, ON: mean=9.833; SD=2.042; hChR2, OFF: mean=9.7; SD=1.856; hChR2, ON: mean=9.085; SD=2.305	two way repeated measures ANOVA; planned comparison between light on and light off for each group; Sidak's multiple comparisons	P=0.815; SS=0.0742, DF=1; MS=0.0742; F(1, 10)=0.0578; adjusted P values for multiple comparisons: control, P=0.808; hChR2, P=0.602
S11F	N=6 control; N=6 hChR2	control, OFF: mean=3141.464; SD=1251.3; control, ON: mean=2359.667; SD=1089.946; hChR2, OFF: mean=2577.152; SD=1481.397; hChR2, ON: mean=1990.271; SD=1597.535	two way repeated measures ANOVA; planned comparison between light on and light off for each group; Sidak's multiple comparisons	P=0.872; SS=16564, DF=1; MS=16564; F(1, 10)=0.0273; adjusted P values for multiple comparisons: control, P=0.523; hChR2, P=0.394
S11G	N=6 control; N=6 hChR2		two way repeated measures ANOVA for each group	control: P>0.999; SS=87.59; DF=799; MS=0.11; F(799, 7990)=0.683; hChR2: P<0.0001; SS=266.7; DF=799; MS=0.334; F(799, 7990)=3.301
S11H	N=6 control; N=6 hChR2	control, OFF: mean=5.111; SD=1.303; control, ON: mean=4.669; SD=0.314; hChR2, OFF: mean=4.1; SD=1.297; hChR2, ON: mean=3.263; SD=1.118	two way repeated measures ANOVA; planned comparison between light on and light off for each group; Sidak's multiple comparisons	P=0.667; SS=0.231, DF=1; MS=0.231; F(1, 10)=0.197; adjusted P values for multiple comparisons: control, P=0.746; hChR2, P=0.379
S11I	N=6 control; N=6 hChR2	control, OFF: mean=1126.27; SD=492.354; control, ON: mean=939.424; SD=245.927; hChR2, OFF: mean=663.804; SD=560.802; hChR2, ON: mean=332.098; SD=352.319	two way repeated measures ANOVA; planned comparison between light on and light off for each group; Sidak's multiple comparisons	P=0.614; SS=31477, DF=1; MS=31477; F(1, 10)=0.271; adjusted P values for multiple comparisons: control, P=0.597; hChR2, P=0.231
S11K	N=10 control; N=11 hChR2		two way repeated measures ANOVA for each group	control: P>0.999; SS=244.8; DF=799; MS=0.306; F(799, 15980)=0.654; hChR2: P<0.0001; SS=336.2; DF=799; MS=0.421; F(799, 15980)=1.347
S11L	N=10 control; N=11 hChR2	control, OFF: mean=7.439; SD=1.521; control, ON: mean=7.812; SD=1.763; hChR2, OFF: mean=8.011; SD=1.636; hChR2, ON: mean=8.814; SD=1.547	two way repeated measures ANOVA; planned comparison between light on and light off for each group; Sidak's multiple comparisons	P=0.151; SS=0.507, DF=1; MS=0.501; F(1, 20)=2.232; adjusted P values for multiple comparisons: control, P=0.156; hChR2, P=0.0016

S11M	N=10 control; N=11 hChR2	control, OFF: mean=1210.406; SD=507.691; control, ON: mean=1353.36; SD=563.168; hChR2, OFF: mean=1366.513; SD=574.794; hChR2, ON: mean=1628.167; SD=562.261	two way repeated measures ANOVA; planned comparison between light on and light off for each group; Sidak's multiple comparisons	P=0.202; SS=38747, DF=1; MS=38747; F(1, 20)=1.742; adjusted P values for multiple comparisons: control, P=0.0708; hChR2, P=0.0011
S12C	N=7 control; N=7 hChR2		two way repeated measures ANOVA for each group	control: P>0.999; SS=1401; DF=39; MS=35.93; F(39, 468)=0.0277; hChR2: P=0.996; SS=27784; DF=39; MS=712.4; F(39, 546)=0.497
S12D	N=6 control; N=6 hChR2	control, OFF: mean=10517.172; SD=1456.715; control, ON: mean=10486.401; SD=1789.038; hChR2, OFF: mean=9777.595; SD=726.394; hChR2, ON: mean=9633.715; SD=1299.517	two way repeated measures ANOVA; planned comparison between light on and light off for each group; Sidak's multiple comparisons	P=0.776; SS=19191, DF=1; MS=19191; F(1, 10)=0.0855; adjusted P values for multiple comparisons: control, P=0.992; hChR2, P=0.848